# Oxygen budget for the north-western Mediterranean deep convection region

Caroline Ulses[1,2], Claude Estournel[1], Marine Fourrier[3], Laurent Coppola[3], Fayçal Kessouri[2,4], Dominique Lefèvre[5], Patrick Marsaleix[1]

[1]Laboratoire d'Etudes en Géophysique et Océanographie Spatiales (LEGOS), Université de Toulouse, CNES, CNRS, IRD, UPS, Toulouse, France
[2]Laboratoire d'Aérologie (LA), Université de Toulouse, CNRS, UPS, Toulouse, France
[3]Sorbonne Université, CNRS, Laboratoire d'Océanographie de Villefranche (LOV), 06230 Villefranche-sur-Mer, France
[4]Southern California Coastal Water Research Project, Costa Mesa, CA, USA
[5]Aix-Marseille Université, Mediterranean Institute of Oceanography (MIO), 13288 Marseille Cedex 9, France

*Correspondence to*: Caroline Ulses (caroline.ulses@legos.obs-mip.fr)

**Abstract.** The north-western Mediterranean deep convection plays a crucial role in the general circulation and biogeochemical cycles of the Mediterranean Sea. The DEWEX (DEnse Water EXperiment) project aimed to better understand this role through an intensive observation platform combined with a modelling framework. We developed a 3 dimensional coupled physical and biogeochemical model to estimate the cycling and budget of dissolved oxygen in the entire north-western Mediterranean deep convection area over the period September 2012 to September 2013. After showing that the simulated dissolved oxygen concentrations are in a good agreement with the in situ data collected from research cruises and Argo floats, we analyze the seasonal cycle of the air-sea oxygen exchanges, as well as physical and biological oxygen fluxes, and we estimate an annual oxygen budget. Our study indicates that the annual air-to-sea fluxes in the deep convection area amounted to 20 mol $m^{-2}$ $yr^{-1}$. 88% of the annual uptake of atmospheric oxygen, i.e. 18 mol $m^{-2}$, occurred during the intense vertical mixing period. The model shows that an amount of 27 mol $m^{-2}$ of oxygen, injected at the sea surface and produced through photosynthesis, was transferred under the euphotic layer, mainly during deep convection. An amount of 20 mol $m^{-2}$ of oxygen was then gradually exported in the aphotic layers to the south and west of the western basin, notably, through the spreading of dense waters recently formed. The decline in the deep convection intensity in this region predicted by the end of the century in recent projections, may have important consequences on the overall uptake of atmospheric oxygen in the Mediterranean Sea and on the oxygen exchanges with the Atlantic Ocean, that appear necessary to better quantify in the context of the expansion of low-oxygen zones.

## 1 Introduction

Deep convection is a key process leading to a massive transfer of oxygen from the atmosphere to the ocean interior (Körtzinger et al., 2004; 2008b; Fröb et al., 2016; Wolf et al., 2018). Its reduction induced by enhanced stratification in response to global warming is one of the primary factors, along with the warming-induced decrease in solubility, slowdown of the overturning circulation, and changes in C:N utilization ratios, that may explain the ongoing decline in open ocean oxygen inventory, or deoxygenation, observed and modelled since the middle of the 20$^{th}$ century (Bopp et al., 2002; Keeling and Garcia, 2002; Keeling et al., 2010; Helm et al., 2011; Andrews et al., 2017; Ito et al., 2017; Schmidtko et al., 2017; Breitburg et al., 2018). The oxygen decline leads to an increase in the volume of hypoxic or even anoxic waters and to the expansion of oxygen minimum zones, which substantially affect life and habitats of the marine ecosystem, and have implications on biogeochemical cycles (Ingall et al., 1994; Levin, 2003; Diaz and Rosenberg, 2008; Breitburg et al., 2009; Naqvi et al., 2010; Stramma et al., 2010; Scholz et al., 2014; Bristow et al., 2017). It is crucial to gain understanding of the actual ventilation occurring in deep convection areas and to continue developing models to predict its future evolution under climate change.

The North-Western (NW) Mediterranean Sea (Fig. 1) is one of the few regions of the world where deep convection takes place (Schott et al., 1996). A cyclonic gyre formed by the Northern Current and the Balearic front leads to the doming of the isopycnals and the rising of high-salinity intermediate waters, the Levantine Intermediate Waters (LIW), close to the surface. In winter, cold and dry northerly winds (Mistral, Tramontane) produce the cooling, evaporation, and a subsequent increase in density of surface waters. The instability of the water column results in the mixing of surface waters with deeper waters, and, when the process is intense, in the formation of new deep waters that spread into the western Mediterranean Sea, such as observed in 2004-2006 (Schroeder et al., 2008a). The depth and horizontal extension of convection in the NW region show a strong interannual variability, driven by both the variability of the winter buoyancy loss and the stratification magnitude prior to the convection period (Mertens and Schott, 1998; Béthoux et al., 2002; Houpert et al., 2016; Somot et al., 2016). The deepest convection takes place in the centre of the Gulf of Lions where the yearly maximum of the mixed layer depth varies from a few hundred meters to 2,500 m when the bottom is reached (MEDOC Group, 1970).

The NW Mediterranean deep convection is responsible for a massive supply of nutrients in the euphotic layer (Severin et al., 2014; Ulses et al., 2016; Kessouri et al., 2017) and an intense bloom in spring when vertical mixing weakens (Bernadello et al., 2012; Lavigne et al., 2013; Auger et al., 2014; Ulses et al., 2016; Kessouri et al., 2018). The formation of deep waters is also at the origin of a huge ventilation of the western Mediterranean Sea (Minas and Bonin, 1988; Copin-Montégut and

Bégovic, 2002; Schroeder et al., 2008a; Schneider et al., 2014; Stöven and Tanhua, 2015; Touratier et al., 2016; Coppola et al., 2017; 2018). Mavropoulou et al.'s (2020) study, based on in situ observations over the period 1960-2011, indicated that its variability is one of the main drivers of the interannual variability of the dissolved oxygen ($O_2$) concentration in the deep waters of the western Mediterranean Sea. Coppola et al. (2018), by analyzing the evolution of observed oxygen profiles in the Ligurian Sea over a 20-year period, suggested that hypoxic conditions (oxygen concentration <2 ml $O_2$ $l^{-1}$ or <61 µmol $O_2$ $kg^{-1}$, Diaz and Rosenberg, 2008; Breitburg et al., 2018) may be reached in water masses at intermediate depths after a period of 25 years without deep convection events (presuming bacterial respiration remains the same). Yet, recently, regional ocean models of the Mediterranean Sea converged to predict a weakening of NW deep convection intensity under climate change scenarios by the end of the 21$^{st}$ century (Soto-Navarro et al., 2020).

One of the objectives of the DEWEX (DEnse Water EXperiment) project carried out in 2012/2013 was to investigate the deep convection process, the formation of North-Western Mediterranean Deep Waters and the impact of deep convection on biogeochemical fluxes (Conan et al., 2018). Three cruises and the deployment of autonomous platforms (glider, Argo floats) provided an unprecedented intensive observation of this region before, during and after a deep convection event, and completed the observation effort during the stratified period operated since 2010 in the framework of the MOOSE-GE (Mediterranean Ocean Observing System for the Environment-Grande Échelle) program (Estournel et al., 2016b). The 2012/2013 event was identified by observational and modelling studies as one of the five most intense deep convection events over the period 1980-2013 (Somot et al., 2016; Herrmann et al., 2017; Coppola et al., 2018) due to extremely strong buoyancy loss (Somot et al., 2016). Regarding the oxygen dynamics, DEWEX winter observations showed a strong increase in the $O_2$ inventory of the entire water column, which was concomitant to the deepening of the mixed layer and was attributed to a rapid intake of atmospheric dissolved oxygen (Coppola et al., 2017). However, these observations remain limited in time and space. Up to date, no high-resolution modelling of the oxygen dynamics in the NW deep convection region that could complete the monitoring effort and provide quantification for the whole area has been yet proposed.

In this study, we take advantage of the DEWEX project to implement and constrain with in situ observations, a 3D coupled physical-biogeochemical model representing the dynamics of dissolved oxygen, and to gain understanding in the variability of the oxygen inventory in the whole NW Mediterranean deep convection area, for the period between September 2012 and September 2013. In this framework, we investigate the seasonal cycle of the oxygen inventory and estimate its annual budget, and we analyze and quantify the relative contribution of air-sea exchanges, as well as of physical and biological processes in the budget. The following document is organized as follows: in Sect. 2, we describe the numerical model, its implementation and the observations used for its assessment. In Sect. 3, we compare our model results with in situ observations. In Sect. 4, we describe the seasonal cycle of atmospheric and physical conditions. In Sect. 5, we examine the seasonal cycle of oxygen inventory and fluxes, as well as the annual oxygen budget. We discuss our results in Sect. 6 and conclude in Sect. 7.

## 2 Material and methods

### 2.1 The numerical model

We use a biogeochemical model forced offline by daily outputs of a 3D hydrodynamic model. Both models and their initial and boundary conditions are described in the following sub-sections.

#### 2.1.1 The hydrodynamic model

The SYMPHONIE model used in this study is a 3D primitive equation model, with a free surface and generalized sigma vertical coordinate, as described in Marsaleix et al. (2008). This model was previously used in the Mediterranean Sea to simulate open-sea convection (Herrmann et al., 2008; Estournel et al., 2016a; Ulses et al., 2016), shelf dense water cascading (Estournel et al., 2005; Ulses et al., 2008b) and continental shelf circulation on the Gulf of Lions shelf (Estournel et al., 2001; 2003; Ulses et al., 2008a). The numerical domain (Fig. 1) covers most of the western Mediterranean basin, using a curvilinear grid with variable horizontal resolution (Bentsen et al., 1999). The mesh size ranges from 0.8 km in the north to 1.4 km in the south. The grid has forty vertical levels with closer spacing near the surface (15 levels in the first 100 m in the center of the convection area characterized by depths of ~2,500 m).

The model was initialized and forced at its lateral boundaries with daily analyses of the configuration PSY2V4R2 based on the NEMO ocean model at a resolution of 1/12° over the Atlantic and the Mediterranean by the Mercator-Ocean International operational system (Lellouche et al., 2013). Following Estournel et al. (2016a), the initial field and open boundary conditions were corrected from stratification biases deduced from comparisons with observations taken during the MOOSE-GE cruise of August 2012. Atmospheric forcing (turbulent fluxes) was calculated using the bulk formulae described by Large and Yeager (2004). Meteorological parameters including downward radiative fluxes were taken from the ECMWF (European Centre for Medium-Range Weather Forecasts) operational forecasts at 1/8° horizontal resolution and 3 hour temporal resolution based on daily analyses. River runoffs were considered based on realistic daily values for French rivers (data provided by Banque Hydro, www.hydro.eaufrance.fr) and Ebro (data provided by SAIH Ebro, www.saihebro.com), and mean annual values for the other rivers.

#### 2.1.2. The biogeochemical model

The biogeochemical model Eco3M-S is a multi-nutrient and multi-plankton functional type model that simulates the dynamics of the biogeochemical decoupled cycles of several biogenic elements (carbon, nitrogen, phosphorus, silicon), and of non-Redfieldian plankton groups (Ulses et al., 2016). The model was previously used to study the biogeochemical processes on the Gulf of Lions shelf (Auger et al., 2011) and in the NW Mediterranean deep convection area (Herrmann et al., 2013; Auger et al., 2014; Ulses et al., 2016; Herrmann et al., 2017; Kessouri et al., 2017; 2018). In this study, the model was extended to describe the dynamics of dissolved oxygen in the ocean interior and the air-sea exchanges of oxygen. Here we only describe the rate of change of the new state variable, the dissolved oxygen concentration, and the parameterization

of the air-sea flux of oxygen, that were included in the model version described in detail by Auger et al. (2011). The rate of change of dissolved oxygen concentration due to biogeochemistry in the water column is governed by the following equation:

$$\frac{dDOx}{dt} = \sum_{i=1}^{3}(GPP_i - RespPhy_i)\gamma_{C/DOx} - \sum_{i=1}^{3}\left(RespZoo_i + RespZoo_i^{add}\right)\gamma_{C/DOx} - RespBac\ \gamma_{C/DOx} - Nitrif\,\gamma_{NH_4/DOx} \quad (1)$$

where $DOx$ is the dissolved oxygen concentration, $GPP_i$ and $RespPhy_i$ are gross primary production and respiration, respectively, for phytoplankton group i; $RespZoo_i$ and $RespZoo_i^{add}$ are basal respiration and additional respiration fluxes to maintain constant N:C and P:C internal ratios, respectively, for zooplankton group i, $RespBac$ is bacterial respiration and $Nitrif$ nitrification. $\gamma_{C/DOc}$ and $\gamma_{NH_4/DOx}$, equal to 1 and 2, respectively, are the mol of $DOx$, used per mol C in respiration and needed to oxidize one mol of ammonium in nitrification as described in Grégoire et al. (2008).

The flux of dissolved oxygen at the air-sea interface, $DOxFlux$, is computed from:

$$DOxFlux = K_w\left(DOx_{sat} - DOx_{surf}\right) \quad (2)$$

where $DOx_{sat}$ and $DOx_{surf}$ (in mmol m$^{-3}$) are the concentration of dissolved oxygen at saturation level and at the surface of the ocean, respectively, and $K_w$ (in m s$^{-1}$) is the gas transfer velocity. The oxygen solubility (or dissolved oxygen at saturation level) is determined using the equation of Garcia and Gordon (1992). The oxygen saturation anomaly (noted $\Delta O_2$) is defined as $\Delta O_2 = (DOx - DOx_{sat})/\,DOx_{sat} \times 100\,\%$. We computed here the gas transfer velocity using the parameterization of Wanninkhof and McGillis (1999) with a cubic dependency to the wind, following the study in the convective Labrador Sea by Körtzinger et al. (2008b) who found that this parameterization was one of those that gave best results and recommended a stronger than quadratic wind speed dependency for high wind speed range. In addition, sensitivity analyses using six various parameterizations of the gas transfer velocity were performed to estimate uncertainties of air-sea exchanges and are discussed in Sect. 6.1. For these sensitivity tests, we used quadratic (Wanninkhof, 1992; Wanninkhof, 2014) and hybrid (Nightingale et al., 2000; Wanninkhof et al., 2009) wind speed dependency parameterizations, as well as parameterizations including air-sea fluxes due to bubbles formation, namely the parameterization proposed by Liang et al. (2013) and the parameterization by Bushinsky and Emerson (2018) who applied in the previous one a multiplicative reduction coefficient of 0.29. To compute the gas transfer velocity, we used the 3 hour wind speed provided by the ECMWF model on a 1/8° grid, in consistency with the hydrodynamic simulation.

Following Kessouri et al. (2017), the biogeochemical model was downscaled from the Mediterranean basin scale to the regional scale used here. The biogeochemical basin scale model was forced by daily fields of temperature, salinity, current and vertical diffusivity from the NEMO model (PSY2V4R2 analyses), which were also used for the boundary conditions of

our hydrodynamic model (Sect. 2.1.1). This basin configuration was initialized in summer 2011, with climatological fields of in situ nutrient concentrations from the oligotrophic period in the Medar/MedAtlas database (Manca et al., 2004) and according to oxygen observations from Meteor M84/3 cruise carried out in April 2011 (Tanhua et al., 2013) and DYFAMED station observations in August 2011 (Coppola et al., 2018). Daily values of the state variables were extracted from the basin-scale run for the initial and lateral boundary conditions of the regional model. This nesting protocol ensures the coherence of the physical and biogeochemical fields at the open boundaries. The regional model was initialized in August 2012. Due to strong vertical diffusivities in the basin scale model, we corrected the initial oxygen concentration, for the north-western region using DYFAMED observations carried out in the summer of 2012 (Coppola et al., 2018), and for the south-western region according to Meteor M84/3 observations. At the Rhone River mouth, nitrate, ammonium, phosphate, silicate and dissolved organic carbon concentrations were prescribed using in situ daily data (P. Raimbault, personal communication). These data, combined with those of Moutin et al. (1998) and Sempéré et al. (2000), were used to estimate dissolved organic phosphorus and nitrogen, and particulate organic matter concentrations as described in Auger et al. (2011). At the other river mouths, climatological values were prescribed according to Ludwig et al. (2010). Dissolved oxygen concentration at the river mouths was set to values at saturation. The deposition of organic and inorganic matter from the atmosphere was neglected in this study. Fluxes of inorganic nutrients and oxygen at the sediment-sea interface were considered by coupling the pelagic model with a simplified version of the meta-model described by Soetaert et al. (2001). The parameters of the latter model were set following the modelling study performed by Pastor et al. (2011) for the Gulf of Lions shelf.

#### 2.1.3. Area of study

For analyses and budget purposes, we defined the deep convection area, as the area where the daily averaged mixed layer depth exceeded 1,000 m at least once during wintertime (red circled area in Fig. 1), according to Kessouri et al. (2017; 2018). It covered an area of 61,720 km$^2$ in 2013. The mixed layer depth is defined as the depth where the potential density exceeds its value at 10 m depth by 0.01 kg m$^{-3}$ (Coppola et al., 2017). Heat fluxes, physical and biogeochemical parameters and fluxes presented in the following sections correspond to values averaged over all model grid points included in this area. The budget of oxygen inventory was computed in two layers based on biogeochemistry processes: in the upper layer (from the surface to 150 m) including the euphotic layer where photosynthesis influences the dynamics of oxygen and in the underlying aphotic layer (from 150 m to the bottom) where only respiration and nitrification processes are taken into account in the model. The maximum depth of the base of the euphotic layer was defined at 150 m, based on the regional minimum value of diffuse attenuation coefficient of light at 490 nm derived from satellite observations (http://marine.copernicus.eu/, products: OCEANCOLOUR_MED_OPTICS_L3_REP_OBSERVATIONS_009_095), and following the studies by Lazzari et al. (2012) and Kessouri et al. (2018).

### 2.2 Observations used for the model assessment

#### 2.2.1 Cruise observations

To assess the horizontal and vertical distribution of the simulated dissolved oxygen concentration, we use in situ observations collected during two cruises carried out in the framework of the DEWEX project on-board the RV *Le Suroît*: the first one, DEWEX Leg1, was carried out during the active phase of deep convection, in February 2013, (Testor, 2013) and the second one, DEWEX Leg2, during the following spring bloom, in April 2013 (Conan, 2013). In addition, we use observations from the 2013 MOOSE-GE cruise, conducted during the stratified, oligotrophic season, in June–July 2013 on-board RV *Tethys II* (Testor et al., 2013). The dissolved oxygen measurements were performed during the DEWEX (Leg1: 74 stations, Leg2: 99 stations) and MOOSE-GE (74 stations) cruises, using a Seabird SBE43 sensor. The calibration and quality control of the measurements were described by Coppola et al. (2017). The accuracy of the measurements was estimated at 2 % of oxygen saturation, i.e. 4 µmol kg$^{-1}$. A Winkler analysis performed on-board was used to adjust the SBE43 raw data, as specified by the GO-SHIP group (http://www.go-ship.org/).

We also compare our model results with high frequency measurements of wind at 10 m and of ocean surface temperature, salinity (thermosalinograph and Conductivity-Temperature-Depth (CTD)) and dissolved oxygen concentration (optode) at 3 m depth using the sea surface water continuous acquisition system (SACES) (Dugenne, 2017) during the two DEWEX cruises.

#### 2.2.2 Argo floats

To evaluate the temporal evolution of the modelled oxygen inventory, we use data of three Argo-$O_2$ floats (floats 6901467, 6901471, 6901487) deployed in the NW Mediterranean Sea during the preconditioning phase (late November 2012) and the active phase (late January 2013) of dense water formation, and operational until the end of the study period (Coppola et al., 2017). Dissolved oxygen measurements were made with a standard CTD sensor, equipped with an oxygen optode with fast time response (Aanderaa 4330). Calibrations of optodes were performed before float and also during the deployment using CTD profiles and seawater samples (Niskin bottles). Details on float deployment strategy and calibration are given by Coppola et al. (2017).

### 2.3 Statistical analysis

In order to quantify the performance of the model in its ability to represent the dynamics of dissolved oxygen for the study period, we computed four complementary metrics following the recommendations of Allen et al. (2007): (1) the standard deviation ratio ($r_\sigma = \frac{\sigma_o}{\sigma_m}$ where $\sigma_m$ and $\sigma_o$ are the standard deviation of model outputs and observations, respectively), (2) the Pearson correlation coefficient $R = \frac{\frac{1}{K}\sum_{k=1}^{K}(y_k^m - \overline{y^m})(y_k^o - \overline{y^o})}{\sigma_m \sigma_o}$ where $K$ is the number of observations, $y_k^m$ is the model output that corresponds to the observation $k$, $y_k^o$, $\overline{y^m}$ and $\overline{y^o}$ are the mean of model outputs and observations respectively; (3) the

normalized root mean square error ($NRMSE = \frac{\frac{1}{K}\sum_{k=1}^{K}\left(y_k^o - y_k^m\right)^2}{\overline{y^o}}$ and (4) the percentage bias ($PB = 100 \times \frac{\overline{y^m} - \overline{y^o}}{\overline{y^o}}$ %). The model results are compared with the observations at the same dates and positions.

## 3 Evaluation of the model

The accurate representation of the winter mixing of water masses is an essential point for the simulation of the dissolved oxygen dynamics in this region, marked by a strong ventilation of the deep waters that plays a crucial role in its seasonal cycle (Copin-Montégut et al., 2002; Touratier et al., 2016; Coppola et al., 2017; 2018). A validation of the hydrodynamic part of the simulation is described by Estournel et al. (2016a), who showed similar spatial distribution of the modelled water column stratification in the entire deep convection area, as well as close modelled time evolution of the temperature profile in the centre of the Gulf of Lions open-sea during the winter, to the observations.

Furthermore, an assessment of the biogeochemical part of the coupled model is presented in Kessouri et al. (2017; 2018). These studies showed that the model is able to accurately reproduce the timing and magnitude of the chlorophyll increase during the spring and autumnal blooms, as well as the concentrations of nutrients and depths of nutriclines and the deep chlorophyll maximum during the stratified, oligotrophic period.

In this study, we focus the evaluation of the coupled model on its ability to realistically represent the dynamics of dissolved oxygen in the deep regions of the NW Mediterranean Sea. For this purpose, first we compare the model results to in situ observations from DEWEX and MOOSE-GE cruises conducted at three key periods: the winter mixing period, the phytoplankton bloom period, and the stratified summer period. Then, we compare the model outputs to Argo data deployed in the area in terms of time evolution of oxygen inventory.

### 3.1 Comparisons to cruise observations

The comparisons of modelled wind velocity and ocean model outputs with in situ observations from the high-frequency SACES are shown in Fig. 2. Modelled wind provided by ECMWF and used to force the hydrodynamic model and to calculate the air-sea oxygen flux is highly correlated with the observations ($R=0.96$, $p$-value$<0.01$). The low values of NRMSE (13.9 %) and percentage bias (-0.5 %) show the accuracy of this variable, found for all ranges of value. Regarding the surface ocean variables, we obtain statistically significant correlations equal to 0.64, 0.83 and 0.83 ($p$-value$<0.01$), between observed and modelled values of respectively surface temperature, salinity, and dissolved oxygen concentration. The NRMSE are equal to 2.0 %, 0.3 % and 5.2 %, respectively. The percentage biases remain negligible for temperature (-0.7 %), salinity (0.002 %) and dissolved oxygen concentration (-1.2 %).

Figures 3 and 4 compare the observed and modelled dissolved oxygen concentration for the stations sampled during the DEWEX and MOOSE-GE cruises, respectively, at the surface (between 5 and 10 m) and along the south-north transect passing across the convection area (stations encircled in black on Fig. 3). Overall, the simulation correctly reproduces the

spatial and temporal variability of the oxygen concentration observed at the surface and in the water column during and between the 3 cruises. During wintertime, the model simulates low surface oxygen concentrations (<220 µmol kg$^{-1}$) in the open sea of the Gulf of Lions and the Ligurian Sea, areas that coincide with the deep vertical mixing regions (Estournel et al., 2016a; Kessouri et al., 2017) (Fig. 3a and 3b). Figure 4a shows the oxygen homogenization of the whole water column between 41.5° N and 42.3° N, the core of the deep convection area. Concentrations above 240 µmol kg$^{-1}$ are modelled in the surface layer on the shelf and in the south at the Balearic front, in accordance with the observations (Fig. 3a, 3b, 4a). The model also agrees with observations showing a layer of low oxygen concentration (minimum concentration <185 µmol kg$^{-1}$ at depths around 500 m) located between 150 m and 1,500 m, mainly in the Levantine Intermediate Water (300-800 m), outside the deep convection area (Fig. 4a). The metrics confirm the good agreement between model outputs and observations with, respectively at the surface and along the south-north transect, a significant spatial correlation of 0.81 and 0.61 (p-value <0.01), a NRMSE of 5.3 % and 15.7 %, and a negligible percentage bias of -1.1 % and 0.01 %.

During the spring cruise, the model represents high dissolved oxygen values (>240 µmol kg$^{-1}$) at the surface throughout the region, as observed (Fig. 3c and 3d). The increase in modelled oxygen concentration in the surface layer between both campaigns is in agreement with observations (Fig. 3a-d and 4a-b). A zone of low oxygen concentration in the intermediate waters is present in the convection area in both datasets (Fig. 4b). However, it is worth noting that this zone of low oxygen concentration is heterogeneous in its magnitude and thickness both in model outputs and in observations, and is not similarly distributed in space in the model compared to the measurements. At the surface and along the transect, the spatial correlation coefficients between modelled and observed dissolved oxygen are equal to 0.59 and 0.30 (p-value<0.01), respectively, the NRMSE to 4.6 % and 19.3 %, respectively, and the percentage biases to -1.2 % and -4.4 %, respectively.

The north-south gradient, with lower surface concentrations in the south of the deep convection area, observed during the stratified period (i.e. MOOSE-GE cruise period in June/July) is then well reproduced by the model (Fig. 3e and 3f). The minimum zone is more established than in spring in both in situ data and model results (Fig. 4c). Both sets of data represent a maximum in the subsurface at depths around 50 m, close to the deep chlorophyll maximum (shown on Fig. 5 in Kessouri et al., 2018), although an underestimation of its magnitude is visible between 41.5° and 42° N in the model (Fig. 4c). We find a spatial correlation coefficient of 0.64 and 0.96 (p-value <0.01), a NRMSE of 3.2 % and 3.5 % and a negligible percentage bias (absolute values ≤0.4 %) between model outputs and observations at the surface and along the north-south transect, respectively.

The metrics computed using all station data from the three cruises are given in Table 1. The modelled dissolved oxygen concentration is significantly correlated with the observed concentration (R≥0.81, p<0.01), in particular for the winter period when the pattern of the oxygen distribution appears to be primarily shaped by deep convection processes, shown to be accurately represented by Estournel et al. (2016a). The model results show low percentage biases (PB <1 %), low NRMSE (<8 %) and standard deviation ratios ranging between 1.13 and 1.35 which indicate a larger variability in the observations than in the model outputs.

### 3.2 Comparison to Argo float data

The model accurately reproduces the magnitude of oxygen inventory in the water column and its time evolution observed using Argo floats during the study period (Fig. 5). The model simulates the increase observed between early December and late February. This increase is estimated at ~20 mol $m^{-2}$ over a layer from the surface to 1,800 m, along the trajectory of the float 6901467 (Fig. 5a), and at ~10 mol $m^{-2}$ over a layer from the surface to 1,000 m, along the trajectory of the float 6901487 (Fig. 5c), both floats being located in the Gulf of Lions at that period. The oxygen inventory remains high during the month of March, and then decreases significantly from early April to early June, in model outputs and Argo observations. In both datasets, the decrease reaches up to 20 mol $m^{-2}$ over 1,800 m along the path of the Argo float 6901471 in the Gulf of Lions (Fig. 5b) and is less pronounced (~10 mol $m^{-2}$) along the trajectory of the float 6901467 in the Balearic Sea (Fig. 5a). More moderate decreases are then simulated and observed until September along all float trajectories. The statistical analysis shows that, in terms of oxygen inventory, significant correlation coefficients are obtained between the model outputs and the 3 float observations ($0.64<R<0.83$, p-value<0.01), NRMSE are smaller or equal to 2.4 % and the absolute values of percentage bias are smaller or equal to 2% (Fig. 5). In terms of dissolved oxygen concentration in the water column, we obtain significant correlation coefficients ($0.56<R<0.93$, p-value<0.01), NRME smaller than 10.5 %, percentage biases smaller than 1% and standard deviation ratios close to 1 for floats 6901471 and 6901487, and of 1.37 for float 6901467 (Table 1).

## 4 Atmospheric and hydrodynamic conditions

In the NW Mediterranean Sea, deep convection takes place every winter but shows a strong interannual variability in its magnitude and spatial extent. This interannual variability is partly related to the variability of heat fluxes (Somot et al., 2016). Over the study period, the convection area was marked by severe heat loss episodes from late October 2012 to mid-March 2013 (Fig. 6a). In particular, there was a first short but intense heat loss event (mean heat flux <-1000 W $m^{-2}$) at the end of October, followed by several long northerly wind episodes when heat loss peaks reached 500 W $m^{-2}$, during the months of December to February (late November to mid-December, mid-January, early February and late February). Finally, a last strong heat loss episode occurred in mid-March after a period of positive heat flux. The wind velocity averaged over the convection period (January 15-March 8, March 15-March 24) was maximum in the centre of the Gulf of Lions, where it reached 10 m $s^{-1}$ (Fig. 7a). From April onwards, the convection region was mainly characterized by heat gains.

In response to the autumnal heat loss events, the mixed layer (ML) began to deepen below 50 m at the end of November (Fig. 6b). Its deepening was strongly enhanced in winter over four periods that coincided with the four episodes of intense northerly wind associated with heat loss mentioned above (Fig. 6a-b). Deep convection reached the bottom layer (~2,000 m) in the core of the convection zone (latitude≈42° N, 4° E<longitude<5° E) in early February and the spatially averaged mixed layer reached a maximum depth of about 1,500 m, at the end of February (Fig. 6b). At the end of the main convection event, end of February/early March the spatially averaged mixed layer abruptly decreased to less than 100 m (Fig. 6b). Finally,

during the secondary convection event from 15 to 24 March, it reached almost 800 m. Figure 7b shows the modelled mixed layer depth (MLD) averaged over the convection periods. It exceeded 1,000 m in a central area of the Gulf of Lions, between 41.5° and 42.5° N and 3.5° and 7° E and was smaller than 500 m in the Ligurian Sea. From mid-April to the end of the period, the mixed layer was shallow (depth <20 m) and its depth remained above the nutriclines (Kessouri et al., 2017) and the deep chlorophyll maximum (Kessouri et al., 2018).

## 5 Results

### 5.1 Seasonal cycle of dissolved oxygen

The good agreement found between model results and in situ measurements (Sect. 3) gave us confidence in the model that we use here to analyze the evolution of oxygen inventory in the deep convection area and to quantify the relative contribution of each oxygen flux in its variation: exchanges at the air-sea interface, as well as physical and biological fluxes in the ocean interior. Based on the evolution of vertical mixing and the phytoplankton growth in the study area, Kessouri et al. (2017) divided the study period into four sub-periods. The first period from September to the end of November, which we will refer to as the autumn period, is characterized by a stratified water column (mean MLD <50 m) and respiration dominating primary production (Kessouri et al., 2018). The second period, from the end of November to the end of March, referred here as the winter period, is characterized by a sustained vertical mixing (mean MLD >50 m). The third period, called spring, ran from late March to early June. It corresponds to the period of restratification of the water column (Estournel et al., 2016a) and of the peak of the phytoplankton bloom at the sea surface followed by the formation of a deep chlorophyll maximum (Kessouri et al., 2018). The last period, summer, from early June to September, is characterized by a strong stratification (mean MLD<20 m) and the permanent presence of a deep chlorophyll maximum below 40 m depth (Kessouri et al., 2018). In the following, we will analyze the dynamics of dissolved oxygen for these four periods. The time evolution of daily oxygen budget terms is shown in Fig. 6d-f, while the time evolution of cumulative oxygen fluxes and the resulting variation in oxygen inventory for the upper (surface-150 m) and deeper (150 m-bottom) layers is presented in Fig. 8. The biological term of the budget is defined as the sum of oxygen production through photosynthesis, and of oxygen consumption through respiration by phytoplankton, zooplankton and bacteria, and through oxidation of ammonium (nitrification) (see Eq. 1). The physical term is decomposed into two transports: a net lateral transport due to advection (positive values correspond to an input for the deep convection area) and a net vertical downward transport at the interface between the two layers, at 150 m depth, due to advection and turbulent mixing. Finally, the time evolution of the dissolved oxygen concentration and the oxygen saturation anomaly, $\Delta O_2$, averaged over the convection area is shown in Fig. 9.

**Autumn** - From September to the end of November 2012 (91 days), respiration exceeded primary production throughout the water column. The net loss in oxygen was maximum in the oxygen minimum zone located at the depths of the Levantine Intermediate Water masses (Fig. 9), formed in the eastern Mediterranean Sea and where the biologically produced, exported

organic matter is progressively remineralized along their path toward the western basin and the Gibraltar Strait. The result of biological processes was a net consumption of oxygen and a decrease of 1.8 mol $m^{-2}$ in oxygen inventory (Fig. 6f and 8). Lateral transport was low for autumn and yielded a slight decrease of 0.9 mol $m^{-2}$ in oxygen inventory (Fig. 8). The heat loss and vertical mixing caused by the northerly wind gust at the end of October 2012 led to a decrease in surface temperature and consequently to an increase in oxygen solubility (Fig. 6c). In addition, the vertical mixing reached the depth of the oxygen maximum present in the subsurface (Fig. 9). This caused its erosion and an increase in the surface oxygen concentration which is, however, lower than the oxygen solubility (Fig. 9 and 6c). From this event, the NW deep convection area became undersaturated at the surface (Fig. 9) and the sea began to absorb atmospheric oxygen (flow towards the ocean of 80 mmol $day^{-1}$ on 29 October, Fig. 6d). Over the autumnal period, the cumulative air-sea oxygen flux amounted to 0.3 mol $m^{-2}$ (Fig. 8a). Globally, the convection area was characterized by a decrease in oxygen inventory of 2.4 mol $m^{-2}$, more than two thirds of which occurred in the upper layer.

**Winter** - The winter period was defined from late November 2012 to late March 2013 but can be further divided into two sub-periods based on the intensity of the vertical mixing (Kessouri et al., 2017). During the first sub-period, from the end of November to mid-January (44 days), the mixing intensified, but remained moderate: the ML averaged over the deep convection area remained above the depth of the maximum euphotic layer (150 m, see Sect. 2.1.3) (Fig. 9). The vertical mixing induced a supply of inorganic nutrients in the upper layer that supported primary production. Kessouri et al (2018) identified the beginning of this period as the beginning of a first bloom. From mid-December, the net biological production of oxygen became positive in the upper layer (Fig. 6f). However, over this sub-period, the influence of biological processes on oxygen inventory remained low (-0.3 mol $m^{-2}$, Fig. 8). Air-to-sea oxygen flux was marked by several peaks, greater than 250 mmol $m^{-2}$ $day^{-1}$ (Fig. 6d), coinciding with cold gales from the north. Its contribution to the oxygen inventory over this sub-period amounted to 3.6 mol $m^{-2}$. Regarding the lateral oxygen export, it contributed to a loss of 1.3 mol $m^{-2}$. The sum of the contributions of the different processes in the water column and at the air-sea interface yielded an increase in $O_2$ inventory of 2.0 mol $m^{-2}$ in the water column. 90 % of this increase occurred in the upper layer, from which 0.7 mol $m^{-2}$ of $O_2$ was exported toward the deeper layers.

The second winter sub-period, from mid-January to late March (69 days), corresponds to the period of deep convection. From the middle to the end of January, the surface water masses previously enriched with oxygen, due to primary production and air-sea exchanges, were mixed with the intermediate water masses characterized by a minimum of oxygen (Fig. 9). From the beginning of February, the vertical mixing intensified, causing a net oxygen transport towards deeper layers (depth > 800 m, Fig. 6e, 8 and 9). $O_2$ concentration decreased significantly at the surface and the difference between surface oxygen concentration and oxygen solubility deepened further, the oxygen saturation anomaly reaching -15% until the end of the convection period (Fig. 9). Over this sub-period, the whole NW convection area was undersaturated at -10% to -15% (Fig. 7c). Strong undersaturation and wind intensity led to very high air-sea fluxes. Several peaks reaching 800 mmol $m^{-2}$ $day^{-1}$ are modelled until mid-March (Fig. 6d). The contribution of air-sea fluxes over this period amounted to 18.0 mol $m^{-2}$ (Fig 8a).

Over the deep convection period, the air-sea oxygen exchanges are characterized by a strong spatial variability (Fig. 7d) with a standard deviation of 38%. The air-to-sea sea oxygen flux averaged over the deep convection period varied between 300 and 460 mmol m$^{-2}$ day$^{-1}$ in the heart of the convection area, and between 65 and 200 mmol m$^{-2}$ day$^{-1}$ in the Ligurian Sea.

With regard to biological processes, as shown in previous studies (Auger et al., 2014; Kessouri et al., 2018), zooplankton growth is largely reduced by the deep convection process due to a dilution induced decoupling of preys and predators. In the upper layer, oxygen production through primary production exceeded oxygen consumption processes (respiration, nitrification). In parallel, the export of organic matter into the intermediate and deep layers during deep convection (Kessouri et al., 2018) led to an increase in remineralization processes and consequently a decrease in oxygen inventory in these aphotic layers. The sum of biological fluxes over the entire water column resulted in a small increase in oxygen inventory of 0.4 mol m$^{-2}$, negligible compared to that induced by air-sea fluxes, in consistency with the previous study of Minas and Bonin (1988).

Over this period, the lateral export of dissolved oxygen had high values, reaching 220 mmol m$^{-2}$ day$^{-1}$ (Fig 6e). In the upper layer, the total lateral transport over the period was low (0.5 mol m$^{-2}$), while it is estimated that in the deeper layers 6.7 mol m$^{-2}$ was exported horizontally from the convection area between mid-February and the end of the convection period (Fig. 8b). The downward transport at the base of the upper layer showed strong peaks reaching 500 mmol m$^{-2}$ day$^{-1}$ (Fig. 6e), concomitant with the peaks of the air-to-sea fluxes and the deepening of the ML.

The model results indicate that atmospheric oxygen injected at the surface and, to a lesser extent, produced by phytoplankton or horizontally advected in the upper layer, was massively transported to the intermediate and deep layers (20.1 mol m$^{-2}$). It is worth noting that vertical fluxes showed a strong spatial variability within the convection area. Over this period, the lateral transport from the aphotic layer outside the convection area represents 33% of the amount of downward transport. Globally, the different contributions led to an increase in the water column oxygen inventory of 12.3 mol m$^{-2}$.

**Spring** (late March to early June, 74 days)- In spring, net biological production of $O_2$ remained high in the upper layer until the bloom peak in mid-April, afterwards it decreased but generally remained positive until the end of that period (Fig. 6f). Oxygen consumption through heterotrophic respiration in the deeper layers also remained relatively high. The result of biological contributions was a small increase of 0.3 mol m$^{-2}$ in the $O_2$ inventory of the water column.

During this period, primary production led to a sharp increase in surface oxygen concentration from 220 to 280 µmol kg$^{-1}$ at the peak of the phytoplankton bloom (Fig 6c), and the latter became above saturation in early April, when the convection area became a source of oxygen for the atmosphere (Fig 6c-d). This oversaturation situation at the surface then persisted until the end of the period. The model simulates significant outgassing during the bloom peak (235 mmol m$^{-2}$ day$^{-1}$ on 18 April 2013, Fig 6d) when the mean saturation anomaly reached a maximum value of 15% (Fig. 6c and 9). Overall, the convection area released 0.8 mol m$^{-2}$ of oxygen to the atmosphere during spring. During this restratification phase, a moderate oxygen export to the deep layers is found (3.2 mol m$^{-2}$, Fig. 8). Lateral export to regions surrounding the convection area continued at a high rate with a cumulative value of 5.1 mol m$^{-2}$. Finally, over this period, the water column

in the convection area was subjected to a 5.7 mol $m^{-2}$ decrease in its oxygen inventory, due to the lateral export of oxygen via the spreading of dense waters in the deeper layers and a slight outgassing to the atmosphere.

**Summer** - During the summer period (87 days), the surface oxygen concentration remained higher than the oxygen solubility (Fig. 6c). A supersaturated situation occurred in the deep chlorophyll maximum zone, due to primary production and a general stratification (Fig. 9). We estimate that the ocean released 1.4 mol $O_2$ $m^{-2}$ to the atmosphere over this period (Fig. 8), mainly during moderate northerly gales. In the whole water column, oxygen-consuming biological processes exceeded primary production overall over this period. The result of biological fluxes was responsible for a consumption of 0.8 mol $m^{-2}$ of oxygen. In addition, the convection area continued to export oxygen to the adjacent zone (1.3 mol $m^{-2}$), but at a lower rate (15 mmol $m^{-2}$ $day^{-1}$) than in the two previous periods (90 mmol $m^{-2}$ $day^{-1}$ over the deep convection period and 69 mmol $m^{-2}$ $day^{-1}$ in spring). Finally, the oxygen inventory decreased by 3.5 mol $m^{-2}$ in the whole water column of the deep convection area (Fig. 8).

### 5.2 Annual oxygen budget

Figure 10 illustrates the oxygen budget in the NW Mediterranean convection area over the period September 2012 to September 2013. At the annual scale, the deep convection area is a net sink of oxygen for the atmosphere, estimated at 20.0 mol $O_2$ $m^{-2}$. 88% (17.7 mol $O_2$ $m^{-2}$) of this amount was injected into the ocean interior during the period when the deep convection process took place.
The annual net biological production of oxygen in the euphotic layer (0-150 m) is estimated at 1.6 mol $O_2$ $m^{-2}$. The net annual NCP (Net Community Production, defined as gross primary production minus community respiration in the euphotic zone) is estimated at 3.9 mol$O_2$ $m^{-2}$ $yr^{-1}$, yielding autotrophy in this area. In the deeper layers (150 m-bottom) an oxygen consumption of 3.8 mol $O_2$ $m^{-2}$ was associated with respiration of heterotrophic organisms by 70% and oxidation of ammonium by 30%. This led to an annual net biological consumption of 2.2 mol $O_2$ $m^{-2}$ over the whole water column.
The model indicates that 27.1 mol $m^{-2}$ of $O_2$ was exported from the upper layer to deeper layers. This net transport toward the bottom occurred for 68% during the events of deep vertical mixing of oxygen-rich surface waters with oxygen-poor underlying waters. Finally, the budget shows that the deep convection area appears as a net source for dissolved oxygen for the rest of the western Mediterranean Sea with an annual net horizontal transport of 15.0 mol $O_2$ $m^{-2}$. This transport breaks down into an input of 5.3 mol $O_2$ $m^{-2}$ in the upper layer, and an export of 20.3 mol $O_2$ $m^{-2}$ in the deeper layer.
At the end of the annual cycle, a negligible decrease (0.3 mol $m^{-2}$ i.e. 0.05%) in the oxygen inventory of the upper euphotic layer is found, while 3.1 mol $m^{-2}$ (i.e. 0.66% of the inventory) were stored in the deeper water masses.

## 6 Discussion

### 6.1 Air-sea oxygen flux

Our model results indicate that the NW Mediterranean deep convection area was a net sink for the atmospheric oxygen at a rate of 20.0 mol m$^{-2}$ yr$^{-1}$ between September 2012 and September 2013, and at a rate of 280 mmol m$^{-2}$ day$^{-1}$ (17.7 mol m$^{-2}$ over 63 days) during the 2013 deep convection period. Inside the area, the annual air-sea flux shows a strong spatial heterogeneity, with a range extending from 2.7 mol m$^{-2}$ yr$^{-1}$ at the periphery to 36.0 mol m$^{-2}$ yr$^{-1}$ in the centre. Considering its sea surface area (61,720 km$^2$), the NW deep convection zone received 1,233 Gmol of oxygen from the atmosphere over the period September 2012 to September 2013, including 1,090 Gmol during the winter 2013 intense vertical mixing period. We showed that the strong oxygen ingassing was essentially driven by a high undersaturation (<-10 %) and intense northerly winds during deep convection period.

Nevertheless, uncertainties in the net uptake rate remain. First, uncertainties are linked to errors in modelled ocean surface variables (dissolved oxygen, temperature and salinity) and wind velocity used for the calculation of the air-sea flux. The comparisons of model results with in situ high-frequency measurements at the surface during the period of maximum flux (deep convection period) indicate a bias of less or close to 1 % and a NRMSE smaller than 14 % for the wind velocity, surface temperature, salinity, and oxygen concentration (Sect. 3.1). A second source of uncertainty is linked to the parameterization chosen for the calculation of the gas transfer velocity. In the standard run, we used the cubic dependence with wind speed parameterization proposed by Wanninkhof and McGillis (1999). Sensitivity analyses were performed using 6 other parameterizations for the calculation of air-sea flux (Wanninkhof, 1992; Nightingale et al., 2000; Wanninkhof et al., 2009; Liang et al., 2013; Wanninkhof, 2014; Bushinsky and Emerson, 2018; see Sect. 2.1.2). Estimates of annual air-sea flux, as well as flux and amount of atmospheric oxygen captured by the study area during the deep convection period, calculated with all these parameterizations, are gathered in Table 2. All estimates show a net sink for atmospheric oxygen for the study area. They range from 14.2 to 20.8 mol m$^{-2}$ yr$^{-1}$ at the annual scale, with a mean value of 16.9 ± 2.7 mol m$^{-2}$ yr$^{-1}$, and from 200 to 290 mmol m$^{-2}$ day$^{-1}$, with a mean value of 240 ± 40 mol m$^{-2}$ day$^{-1}$, during the deep convection. Both estimates in the standard run are in the upper range of all estimates. Considering all estimates, we determine an uncertainty (standard deviation) of 16% for the annual and convection period air-sea flux. This uncertainty, associated with the parameterization of the gas transfer velocity, propagates to the estimates of vertical and lateral transport of oxygen in the ocean interior. Depending on the gas transfer parameterization used, at the annual scale, downward export below the euphotic zone ranges from 22.2 to 27.6 mol m$^{-2}$ yr$^{-1}$ (mean value: 24.6 ± 2.1 mol m$^{-2}$ yr$^{-1}$), lateral transport from 5.0 to 6.0 mol m$^{-2}$ yr$^{-1}$ (mean value: 5.6 ± 0.4 mol m$^{-2}$ yr$^{-1}$) in the euphotic layer, and from -17.3 to -20.5 mol m$^{-2}$ yr$^{-1}$ (mean value: -18.6 ± 1.2 mol m$^{-2}$ yr$^{-1}$) in the aphotic layer. During the deep convection event, downward export below the euphotic zone ranges from 14.3 to 18.7 mol m$^{-2}$ yr$^{-1}$ (mean value: 16.2 ± 1.8 mol m$^{-2}$ yr$^{-1}$). The uncertainty on the transport terms of the annual budget thus remains smaller than 10%. The values of the NRMSE between cruise observations and modelled dissolved oxygen from sensitivity tests are found very close to the NRMSE obtained for the standard run. Slightly smaller

NRMSE (~10%) are found only for the winter DEWEX-Leg1 cruise period using the parameterizations of Wanninkhof and McGillis (1999) and Liang et al. (2013), which give a higher transfer coefficient than the other parameterizations.

Previous studies based on in situ observations have proposed estimates for the air-sea oxygen flux in the study area. Our modelled seasonal cycle of air-sea oxygen flux agrees with the results of Copin-Montégut and Bégovic (2002) and Coppola et al. (2018) in the Ligurian Sea, at the DYFAMED site, who observed an annual cycle with a net ingassing from December to March and net outgassing from April to November. In the Ligurian Sea the deep convection process does not occur each winter. When occurring, it is generally shorter and shallower than in the centre of the Gulf of Lions, the core of dense water formation. Coppola et al. (2018) using temperature, salinity and oxygen monthly profiles and the gas transfer parameterization of Ho et al. (2006), estimated for the period 1994-2014 a monthly air-to-sea flux varying from -15.1 to 14.8 mol $O_2$ $m^{-2}$ $yr^{-1}$, with an annual mean value of -2.6 mol $O_2$ $m^{-2}$ $yr^{-1}$. Over this 20-year period, the authors identified one winter, winter 2005/2006, with intense vertical mixing reaching the deep layers, and four winters (1999, 2000, 2006 and 2013) with moderate vertical mixing reaching intermediate depths. From the difference in $O_2$ inventory between December 2005 and April 2006, they deduced that 24 mol $m^{-2}$ of atmospheric $O_2$ were injected between 350 and 2000 m at a rate of 300 mmol $m^{-2}$ $day^{-1}$. At the same site, Copin-Montégut and Bégovic (2002) estimated an air-sea ingassing of 5 and 2.6 mol $O_2$ $m^{-2}$ respectively for the moderate cold winters 1999 (for 26 days, rate of 190 mmol $m^{-2}$ $day^{-1}$) and 2000 (for 23 days, rate of 110 mmol $m^{-2}$ $day^{-1}$) respectively, using in situ surface measurements of oxygen in winter, and the formulation of gas transfer velocity from Wanninkhof and McGillis (1999). Those estimates were twice as small as their observation of variation in the oxygen content in the first 600 m depth, namely 11 and 15 mol $m^{-2}$ (in one month) for winters 1999 and 2000 respectively. Those authors suggested an underestimation in their estimates due to low-frequency measurements and an underestimation of the gas transfer coefficient. At the same location, for the study period, we found a net oxygen ingassing of 9.2 mol $m^{-2}$ $yr^{-1}$ at the annual scale, and 135 mmol $O_2$ $m^{-2}$ $day^{-1}$ during the period of deep convection events (63 days) (Fig. 7d). Thus our calculation of atmospheric oxygen uptake in the Ligurian Sea is close to the ones of Copin-Montégut and Bégovic (2002) for moderate convective winters. Our estimates in the centre of the Gulf of Lions, where convection reached the deep waters, with value of 20-28 mol $O_2$ $m^{-2}$ during deep convection are also close to the estimate by Coppola et al. (2018) for the intense vertical mixing winter 2005/2006 in the Ligurian Sea.

Finally, our model calculation of air-sea oxygen flux for the NW Mediterranean is in the same range found for other worldwide deep convection areas. At the centre of the Labrador Sea, Körtzinger et al. (2008b) found an annual air-sea ingassing of 10.0 ± 3.1 mol$O_2$ $m^{-2}$ $yr^{-1}$ over the period 2004/2005, using in situ observations at the K1 mooring site and the Wanninkhof (1992) parameterization. By quantifying the relative contribution of biological and lateral fluxes, Koelling et al. (2017) estimated an oxygen ingassing of 29.1 ± 3.8 mol $m^{-2}$ over the winter 2014/2015 at the same mooring site K1. Wolf et al (2018) derived from Argo float observations in the Labrador Sea mean air-sea fluxes with a large range of values varying from 5.7 to 22.8 mol $m^{-2}$ $yr^{-1}$ using various parameterizations. Using bubble parameterizations (Liang et al., 2013; Yang et

al., 2017), their estimates of atmospheric oxygen uptake ranged from 21.6 to 36.6 mol $m^{-2}$ for the convective winter 2003/2004. For the Irminger Sea, Maze et al. (2012) estimated an abiotic air-sea oxygen flux of 13 ± 3 mol $m^{-2}$ $yr^{-1}$ for the years 2002, 2004 and 2006, using an optimization method and observations from three surveys and Word Ocean Atlas 2009.

### 6.2 The role of the NW deep convection area in the ventilation of the western Mediterranean Sea

Open-sea convection and shelf dense water cascading (Canals et al, 2006; Ulses et al., 2008b) in the NW Mediterranean are the main mechanisms for the ventilation of the entire western Mediterranean Sea. Over the past decades, several observational studies reported increases in $O_2$ concentration in deep water masses at several sites in the western Mediterranean where deep convection did not occur and where winter vertical mixing was limited to surface or intermediate levels. Coppola et al. (2018) associated the high concentrations of $O_2$ observed in deep layers of the Ligurian Sea in 1994 and 2005, when convection was limited to intermediate waters, with the arrival of deep water formed in the open-sea of the Gulf of Lions or formed on the Gulf of Lions shelf and cascading down to the deep basin. Using measurements from 5 cruises, Schroeder et al. (2008a) documented an abrupt increase in heat, salt and $O_2$ inventory of deep waters in an extensive area of the western Mediterranean, occurring in 2005 and 2006. The authors attributed these changes, referred as the Western Mediterranean Transient (hereafter WMT - CIESM, 2009), to the propagation of the new dense waters formed in the NW deep convection area during the winters 2004/2005 and 2005/2006, when severe weather conditions caused intense dense water formation (Lopez-Jurado et al., 2005; Schroeder et al., 2006). The study of Schroeder et al. (2008a) showed the presence of these new $O_2$-rich deep waters in the Balearic Sea, the Ligurian Sea and in the Algerian sub-basin in June 2005, and their propagation to the whole Algerian sub-basin and the west of the Alboran Sea in October 2006. New oxygenated waters were also observed in the entire deep layers of the Algerian sub-basin in 2011 (Schneider et al., 2014; Stöven and Tanhua, 2015) and 2014 (Keraghel et al., 2020), but were not yet detected in the Tyrrhenian Sea in 2011 (Schneider et al., 2014; Stöven and Tanhua, 2015).

Somot et al. (2016) found that winter 2012/2013 is one the five winters over the 33 year period 1980-2013 showing high dense water formation rates (above 0.6 Sv), using the CNRM-RCSM4 model. According to their estimates, the cumulative volume of dense water formed over the winters 2011/2012 (0.45 Sv) and 2012/2013 (0.7 Sv), amounting to 1.15 Sv, may be close to the volume of dense water formed in winter 2004/2005 of 1.2 Sv. As a result, these successive 2012 and 2013 deep convection events could have been responsible for a similar ventilation as the one observed after the event of 2005 (Schroeder et al., 2008b; Schneider et al., 2004; Stoven and Tanhua, 2015), assuming similar air-sea exchanges.

Our modelling study indicates that, over the period September 2012 to September 2013, the upper layer of the NW deep convection area captured 5.3 mol $O_2$ $m^{-2}$ from the surrounding regions, in addition to the 20.0 mol $m^{-2}$ of oxygen from the atmosphere, while the deeper layers released 20.3 mol $m^{-2}$ toward the adjacent seas (Sect. 5.2). Considering the deep convection surface area of 61,720 $km^2$, the lateral transport led to a gain in the upper layer of 330 Gmol $yr^{-1}$ in the convection area and a loss of 1,250 Gmol $yr^{-1}$ towards the adjacent deep areas. As a result, the NW convection area appears

as a source of 920 Gmol yr$^{-1}$ of oxygen for the rest of the western basin for the period 2012/2013. Our model outputs show that the $O_2$-rich dense waters formed in the NW deep convection area propagated towards the Balearic Sea, first at intermediate depths (150 m-800 m) from the beginning of the winter mixing period, and then through deep layers (800 m-bottom) from mid-February (not shown). These water masses flowed then mostly towards the south of the western basin, while a smaller part was advected back in the convection area through meso-scale circulations counteracting the effect of the intrusions of low oxygen Levantine Intermediate Water during the restratification period, in increasing the oxygen inventory of intermediate waters (not shown). A preferential pathway to the south of the basin was the one along the eastern coast of Minorca in the Algerian sub-basin, in agreement with previous observational and modelling studies who examined the spreading of waters formed in winter in the NW region (Pinot and Ganachaud, 1999; Schroeder et al 2008b; Beuvier et al., 2012). Our simulated circulation of oxygen in the western basin is also consistent with the study of Piñeiro et al. (2019) who reported the arrival of new dense water masses formed in the deep convection area east off Minorca over the 2011–2013 period using temperature and salinity observations at the hydrographic stations RADMED. In our model outputs, the offshore Balearic Sea (bathymetry >1,000 m, surface area: 19,700 km$^2$) and Algerian sub-basin (bathymetry >1,000 m, surface area: 171,610 km$^2$) experienced an increase in their oxygen inventory, during and after the NW deep convection events, receiving oxygen through lateral transport (271 Gmol and 1,276 Gmol, respectively) while the amounts of oxygen captured at the air-sea interface during the period of intense vertical mixing were smaller in those areas than in the NW deep convection area by a factor of 10 and 3 respectively (i.e. 104 Gmol and 385 Gmol versus 1,090 Gmol for the NW deep convection area). This suggests that an important part of the oxygen absorbed at the air-sea interface of the NW deep convection area, exported first vertically towards its deeper layers and then horizontally towards the adjacent regions, was stored, at least temporarily, in the Algerian sub-basin.

Finally, our results demonstrate that the total oxygen supply by air-sea exchanges in the NW deep convection region for the period 2012/2013 (1,233 Gmol yr$^{-1}$), which was then mainly released to adjacent seas in the aphotic layer, constitutes a major source of oxygen at the scale of the whole Mediterranean Sea. Indeed this supply is close to the biological oxygen consumption within the Mediterranean Sea estimated at 1,545 Gmol yr$^{-1}$ by Huertas et al. (2009) using in situ measurements at the Strait of Gibraltar over the period 2005/2007.

The present study of the period 2012/2013 constitutes a first step in our analysis and quantification of the oxygen budget for the western Mediterranean Sea. Previous observational studies (Coppola et al., 2018; Mavropoulou et al., 2020) over periods of 20 years or more, showed that the mean oxygen concentration in the western Mediterranean and in particular in the NW deep convection area is subjected to a strong interannual variability, mainly in response to the variability of deep convection, the latter being influenced by transient changes as the WMT event. A deeper analysis of the spreading of the oxygen enriched dense waters, formed in the NW deep convection area, in the western basin and toward the Atlantic Ocean through the Gibraltar Strait will be conducted in a further study using a numerical simulation with extended domain and period.

### 6.3 Net community production

Our budget calculation shows that in this region characterized by intense vertical mixing the biological terms remained very low compared to the air-sea oxygen flux over the period 2012/2013. Our modelling results indicate that the net biological production of oxygen in the euphotic layer was positive from mid-December to the end of July and negative the rest of the year. It was maximum during the spring bloom from mid-March to mid-April. We estimate a net annual NCP (in the upper layer) of 46.8 gC $m^{-2}$ $yr^{-1}$ (3.9 mol$O_2$ $m^{-2}$ $yr^{-1}$, see Eq. 1). This indicates a net autotrophy for the euphotic layer of the NW Mediterranean deep convection area over the period September 2012-September 2013. If consumption of oxygen through nitrification is considered, net biological production amounted to 1.6 mol$O_2$ $m^{-2}$ $yr^{-1}$. It is worth noting that nitrification, discussed in Kessouri et al. (2017) who estimated a nitrogen budget using the same coupled model, accounts only for 7% of the total oxygen consumption but for 60 % of the NCP, suggesting that this process should be considered when estimating NCP from oxygen concentration.

Our value of NCP is higher than the net downward export of organic carbon at the base of the euphotic layer estimated at 35 gC $m^{-2}$ $yr^{-1}$ (25 gC $m^{-2}$ $yr^{-1}$ for particulate organic carbon and 10 gC $m^{-2}$ $yr^{-1}$ for dissolved organic carbon) by Kessouri et al. (2018) over the same period and using the same coupled model. Also using the same coupled model, Kessouri et al. (2017), who analyzed the nitrogen cycle over the study period, obtained a new primary production varying from 65 to 77 gC $m^{-2}$ $yr^{-1}$ in the deep convection zone. By analyzing the carbon, nitrogen and oxygen cycles, the NW deep convection region is always found to be an autotrophic ecosystem. The discrepancies in magnitude obtained depending on the element considered reflect different dynamics for these elements in the euphotic layer, possibly due to variable $O_2$:C:N ratios in biological production and consumption processes as shown by Copin-Montégut (2000) in the Ligurian Sea using high-frequency measurements.

Our estimate of NCP is smaller than the estimate of 85.2 gC $m^{-2}$ $yr^{-1}$ in the Ligurian Sea at the DYFAMED site over the period 1994-2014 by Coppola et al. (2018) using monthly observations. It is close to the estimate by Ulses et al. (2016) of 42.8 gC $m^{-2}$ $yr^{-1}$ over the period 2003-2008 using the same numerical model and considering an area extending to the whole offshore NW Mediterranean and a 100 m thick upper layer. It is also similar to the previous estimates of new primary production by Severin et al. (2014) varying from 46 to 63 gC $m^{-2}$ $yr^{-1}$ over the period February/March 2011, based on in situ nutrient concentrations.

NCP is often used to estimate the strength of the biological pump and the potential capacity of a system to capture atmospheric $CO_2$. Although the NW deep convection pelagic ecosystem appears as a net annual sink for atmospheric $CO_2$ from our modelling results and previous studies (Coppola et al., 2018; Ulses et al., 2016), the role of this region in terms of carbon sequestration remains highly uncertain. Deep convection generates a strong downward transport of organic carbon below the euphotic layer (Ulses et al., 2016; Kessouri et al., 2018). A large amount of organic carbon transferred below the euphotic zone is then consumed and remineralized after deep convection (Santinelli et al., 2010) leading to an increase in $CO_2$ inventory into the deeper reservoir that could be raised back in the euphotic layer during the following deep convection

events as shown in the Atlantic Ocean by Körtzinger et al. (2008a) and in the Pacific Ocean by Palevsky et al. (2016). Episodes of oversaturation of sea surface $pCO_2$ related to atmospheric $pCO_2$ were reported during short wind gusts and intense vertical mixing events in the Ligurian Sea (Copin-Montégut et al., 2004; Merlivat et al., 2018) and in the central Gulf of Lions open-sea (Touratier et al., 2016). The authors explained those oversaturation episodes by the increase in $CO_2$ concentration at the ocean surface induced by the mixing of surface $CO_2$-poorer waters with deep $CO_2$-rich waters. Those punctual observations suggested short releases of $CO_2$ by the ocean induced by deep convection. On the other hand, using a 0.5° resolution array of 1D hydrodynamic/biogeochemical coupled models of the upper layer in the Mediterranean Sea over the period 1998-2004, D'Ortenzio et al. (2008) estimated that the NW region is a sink for atmospheric $CO_2$ in winter and at the annual scale (between 12 to 24 gC $m^{-2}$ $yr^{-1}$) and found that biological processes dominate air-sea exchanges and mixing processes in this region most of the year. In another region of deep convection, the Labrador Sea, DeGrandpre et al. (2006) and Körtzinger et al. (2008b) also found an annual uptake of atmospheric $CO_2$ that amounted to respectively 55.2 gC $m^{-2}$ $yr^{-1}$ for the period 2000-2001, and 32.4 ± 9.6 gC $m^{-2}$ $yr^{-1}$ for the period 2004-2005, using mooring observations (and also a 1D biogeochemical model for DeGrandpre et al. (2006)).

In the study area, our results show that lateral transport dominated the budget of oxygen during the restratification period when deep dense waters spread in the western basin and LIW reintegrated the deep convection zone. This suggests that 3D biogeochemical-physical coupled models, including a carbonate system module, could be useful tools to complete the previous 1D studies of dissolved inorganic carbon dynamics and integrate on an annual scale the exchanges at the air-sea interface by taking into account lateral transport and meso-scale structures influencing the spreading of water masses and their compounds during convection and restratification phases and impacting the budgets.

## 7 Conclusion

Our study is the first attempt to describe the seasonal cycle of dissolved oxygen and to estimate the oxygen budget over the whole NW Mediterranean deep convection area, using a high-resolution 3D coupled physical-biogeochemical model. The assessment of the model results using in situ measurements from DEWEX and MOOSE-GE cruises and from Argo-$O_2$ floats shows the ability of the model to capture the main spatial and temporal variability of dissolved oxygen observed. From our modelling results, the following conclusions can be drawn for the period 2012/2013:

- The seasonal cycle of surface dissolved oxygen in this area exhibited a winter period with strong undersaturation due to a decrease in temperature and surface dissolved oxygen concentration, induced by strong heat loss and vertical mixing of surface $O_2$-rich waters with the underlying $O_2$-low waters. The undersaturation averaged over the whole area indeed reached -15% during the deep convection event. During the stratified period, an oversaturation situation occurred with a maximum surface value of 15% during the peak of the spring bloom.
- The NW Mediterranean deep convection area acted as a large sink for atmospheric oxygen. We estimate that the area captured 20 mol $m^{-2}$ $yr^{-1}$ of atmospheric oxygen at the annual timescale. An uptake of 18 mol $m^{-2}$ of

atmospheric oxygen, which equals to 88% of the annual uptake, took place during the deep convection period. This uptake is characterized by a high spatial variability, with a standard deviation of 38% in this area including the open-sea of the Gulf of Lions and Ligurian Sea. The magnitude of the uptake is maximum inside a central zone of the Gulf of Lions where the average over the deep convection period (63 days) reached a rate ranging between 300 and 460 mmol $O_2$ $m^{-2}$ $day^{-1}$.

- The NW Mediterranean deep convection area represents a major source of oxygen for the intermediate and deep western Mediterranean Sea. Based on the rate of dense water formation (Somot et al., 2016), the ventilation due to deep convection in the NW area in 2012/2013 may represent half of the ventilation observed in 2004/2005 by Schroeder et al. (2008a). The magnitude of atmospheric $O_2$ uptake and lateral transport to the adjacent regions in the aphotic layers in 2012/2013 is close to the magnitude of the oxygen consumption of the whole Mediterranean Sea estimated by Huertas et al. (2009).
- Sensitivity tests to the parameterization of the gas transfer velocity yield an estimate of the budget terms (air-sea exchanges and transport terms) uncertainty of 10-16%.
- As expected for this very energetic region, the annual budget of oxygen is clearly dominated by air-sea exchanges and physical transport over convective years such as 2012/2013. The net biological production in the euphotic zone is estimated to account for 10%, i.e. 2 mol $O_2$ $m^{-2}$ $yr^{-1}$, of the net atmospheric oxygen uptake. In deeper depths, heterotrophic organisms' respiration and nitrification resulted in an oxygen consumption of 4 mol $m^{-2}$ $yr^{-1}$.
- The NW Mediterranean deep convection area is found to be an autotrophic ecosystem with an annual NCP (in the 150 m upper layer) estimated at 47 gC $m^{-2}$ $yr^{-1}$.

The high interannual variability of deep convection in the NW Mediterranean (Houpert et al., 2016; Somot et al., 2016) suggests a high variability of the oxygen budget. Further modelling works at pluri-annual and multi-decadal scales are thus needed to investigate the interannual variability of the annual budget over the whole western basin, as well as the evolution of this budget under climate warming, the effects of which could have been masked for the time being by the significant impacts of climatic transient shifts such as WMT according to Mavropoulou et al. (2020).

**Date availability**

The Argo data are available on the Coriolis platform (http://www.coriolis.eu.org), MOOSE data on SEANOE/SISMER (https://www.ir-ilico.fr/en/Data-access/MOOSE) and DEWEX data on the Mermex database (https://mistrals.sedoo.fr/MERMeX/). Results of simulations are available on request (caroline.ulses@legos.obs-mip.fr).

**Competing interest**

The authors declare that they have no conflict of interest.

**Acknowledgments**

This study is a contribution to the MerMex (Marine Ecosystem Response in the Mediterranean Experiment) project of the MISTRALS international program. The numerical simulations were performed using the SYMPHONIE model, developed by the SIROCCO group (https://sirocco.obs-mip.fr/), and computed on the cluster of Laboratoire d'Aérologie and HPC resources from CALMIP grants (P1325, P09115 and P1331). We acknowledge the scientists and crews of the Flotte océanographique française (https://www.flotteoceanographique.fr/) who contributed to the cruises carried out in the framework of the DEWEX project and MOOSE program (CNRS-INSU).

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

**Table 1.** Statistical analysis of model results: Pearson correlation coefficient (R), normalized root mean square error (NRMSE), percentage bias (PB) and standard deviation ratio, calculated between modelled dissolved oxygen concentrations and observations from DEWEX winter and spring cruises and MOOSE-GE summer cruise, and from Argo-$O_2$ platforms.

| | **R** | **NRMSE, %** | **PB, %** | **Std ratio** |
|---|---|---|---|---|
| **DEWEX Leg1** | 0.86 (p<0.01, n=2960) | 5.6 | -0.59 | 1.34 |
| **DEWEX Leg2** | 0.93 (p<0.01, n=3960) | 4.9 | 0.55 | 1.13 |
| **MOOSE 2013** | 0.81 (p<0.01, n=2960) | 7.6 | 0.51 | 1.35 |
| **Float 6901467** | 0.56 (p<0.01, n=5120) | 10.3 | -0.12 | 1.37 |
| **Float 6901471** | 0.93 (p<0.01, n=4480) | 3.0 | -0.19 | 0.99 |
| **Float 6901487** | 0.88 (p<0.01, n=4720) | 4.1 | -0.11 | 1.01 |

**Table 2.** Estimate of air-to-sea oxygen flux ($F_{A-S}$) for the period September 2012-September 2013 and during the deep convection (January 15-March 8, March 15-March 24), using different parameterizations of gas transfer velocity.

| **Gas exchange parameterization** | **Annual $F_{A-S}$ $molO_2\ m^{-2}\ yr^{-1}$** | **$F_{A-S}$ and *amount exchanged at the air-sea interface* during the 2013 deep convection event $mmolO_2\ m^{-2}\ day^{-1}$ - *$molO_2\ m^{-2}$*** |
|---|---|---|
| **Wanninkhof and McGillis (1999) (used in the standard run)** | 20 | 280 - *18* |
| **Wanninkhof et al. (1992)** | 18 | 260 - *16* |
| **Nightingale et al. (2000)** | 15 | 210 - *13* |
| **Wanninkhof et al. (2009)** | 15 | 220 - *13* |
| **Liang et al. (2013)** | 21 | 290 - *18* |
| **Wanninkhof et al. (2014)** | 15 | 220 - *14* |
| **Bushinsky and Emerson (2018)** | 14 | 200 - *12* |
| **Mean (standard deviation)** | 17 (3) | 240 (40) - *15 (2)* |

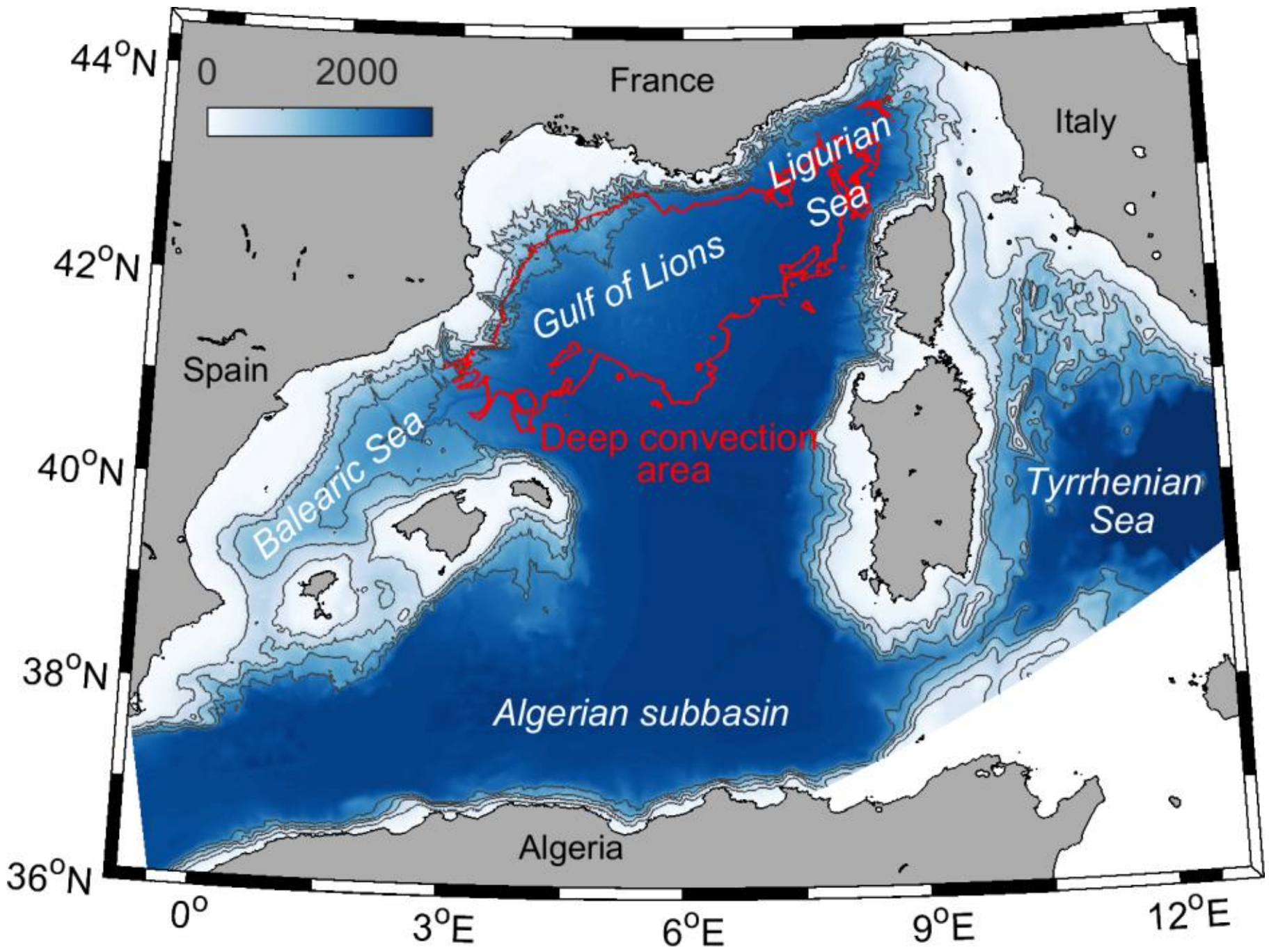

**Figure 1.** Model domain and bathymetry (m) in the western Mediterranean Sea. The area circled in red corresponds to the area of study, the deep convection area.

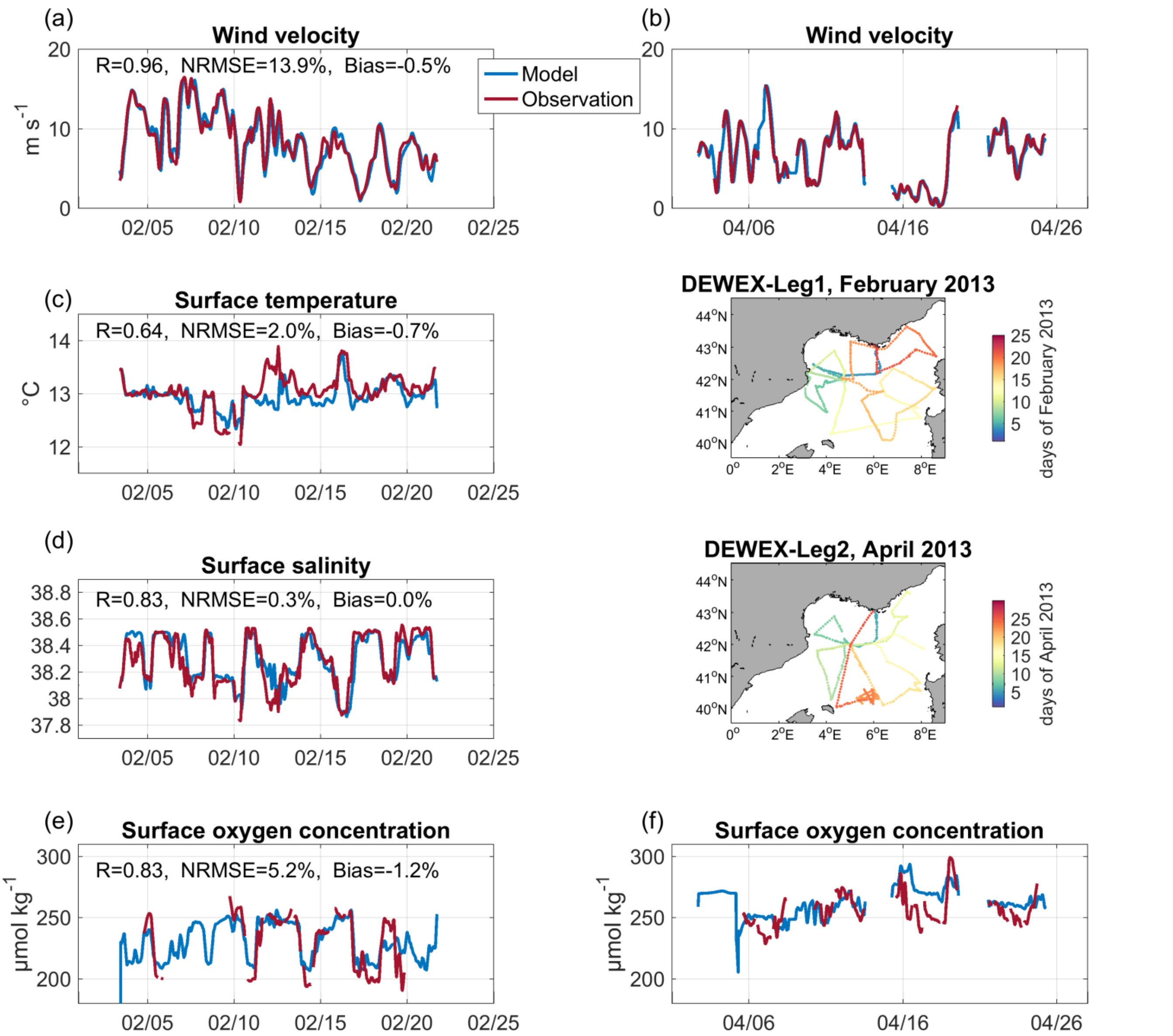

**Figure 2.** Time evolution during DEWEX Leg1 (in February 2013, left column) and Leg 2 (in April 2013, right column) cruises of observed (red) and modelled (blue) (a,b) wind velocity (m $s^{-1}$), and surface (c) temperature (°C), (d) salinity and (e,f) dissolved oxygen concentration (µmol $kg^{-1}$). Trajectories of the measurements during DEWEX Leg1 and Leg2 cruises are indicated on inserted maps. Modelled wind velocity was provided by ECMWF. No surface temperature and salinity data is available over the period of DEWEX Leg2. The metrics indicated for the modelled wind velocity and surface oxygen concentration were calculated for both DEWEX Leg 1 and Leg 2
periods.

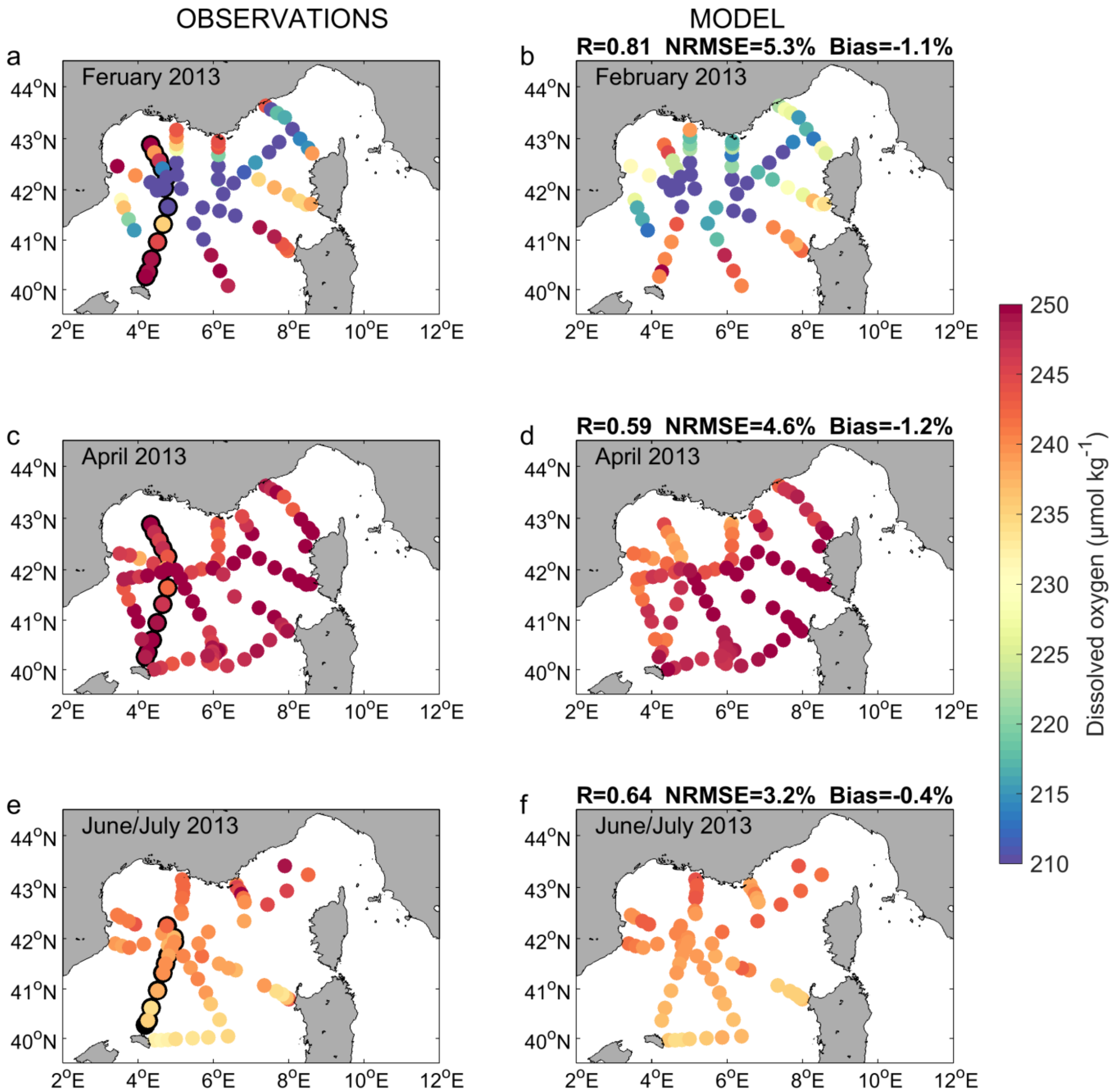

**Figure 3.** Surface dissolved oxygen concentration (µmol kg$^{-1}$) observed (left) and modelled (right) over the (a,b) DEWEX Leg1 (1-21 February 2013), (c,d) DEWEX Leg2 (5-24 April 2013) and (e,f) MOOSE-GE (11 June-9 July 2013) cruise periods. The black-circled dots correspond to the measurement stations shown in Fig. 4.

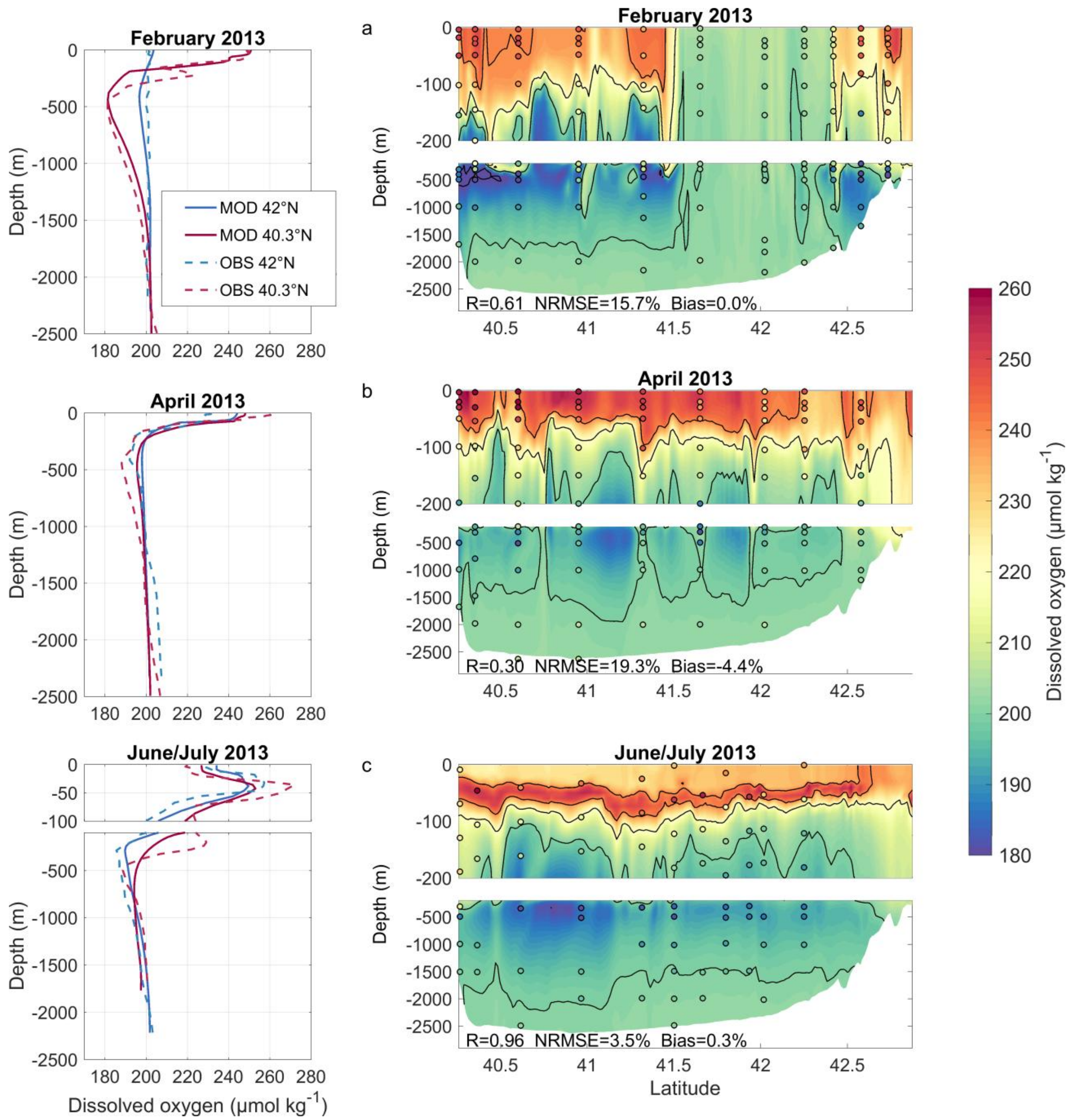

**Figure 4.** Comparison between model outputs and observations on a transect crossing the deep convection area (stations are circled in black on Fig. 3a,c,e) over the (a,b) DEWEX (Leg 1: 10-12 February 2013, Leg2: 8-10 April) and (c) MOOSE-GE (27 June-5 July 2013) cruise periods. Left: Observed and modelled profiles at 42° and 40.3° N. Right: vertical section of dissolved oxygen concentration (µmol kg$^{-1}$) along the transect; the model is represented by background colors and observations are indicated in colored circles.

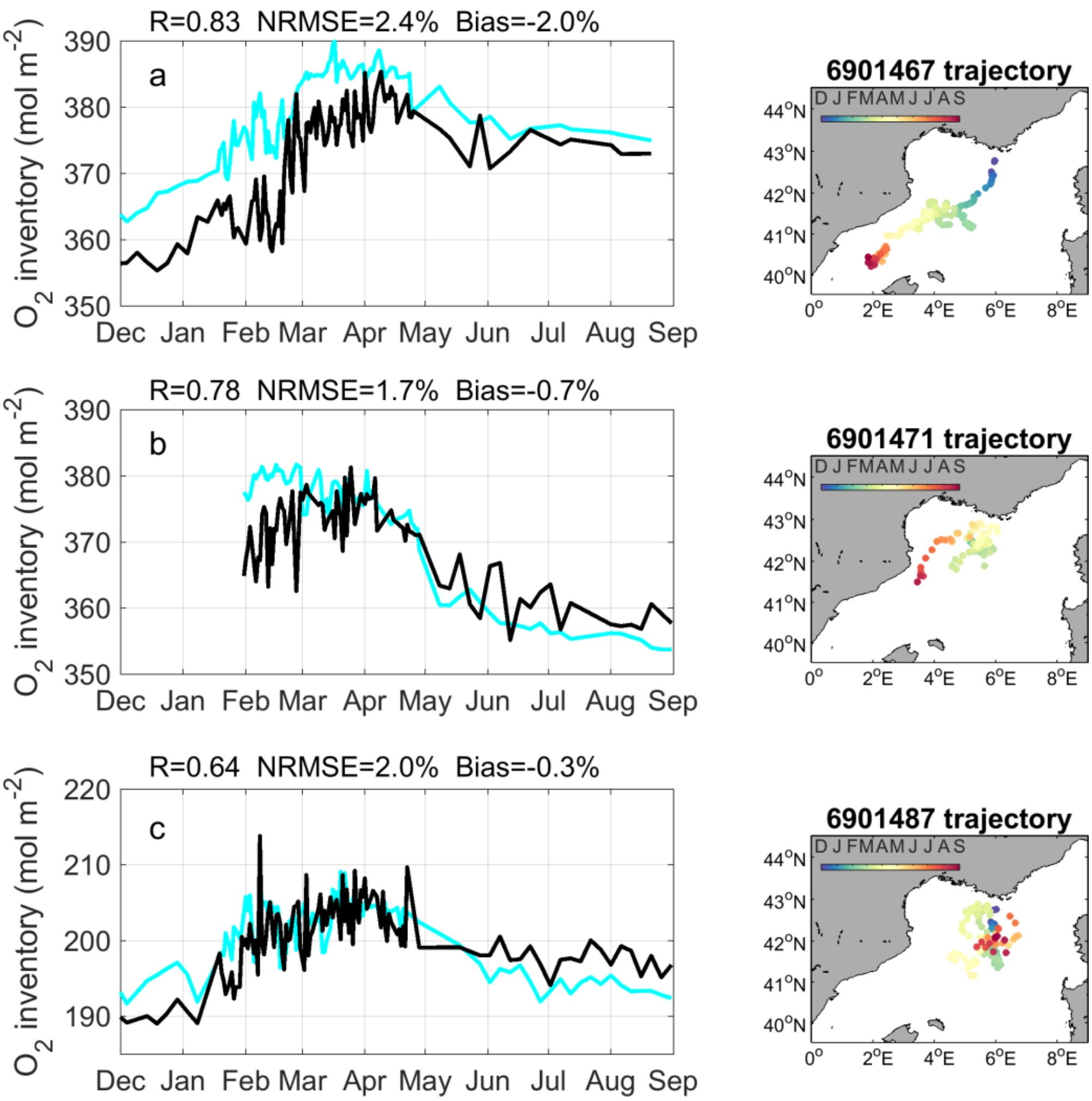

**Figure 5.** Left: oxygen inventory integrated from the surface to 1,800 m depth (a,b) or 1,000 m (c) (mol $m^{-2}$) in Argo float measurements (cyan) and model outputs (black) along Argo (a) 6901467, (b) 6901471, and (c) 6901487 float trajectories. Right: trajectories of corresponding Argo float.

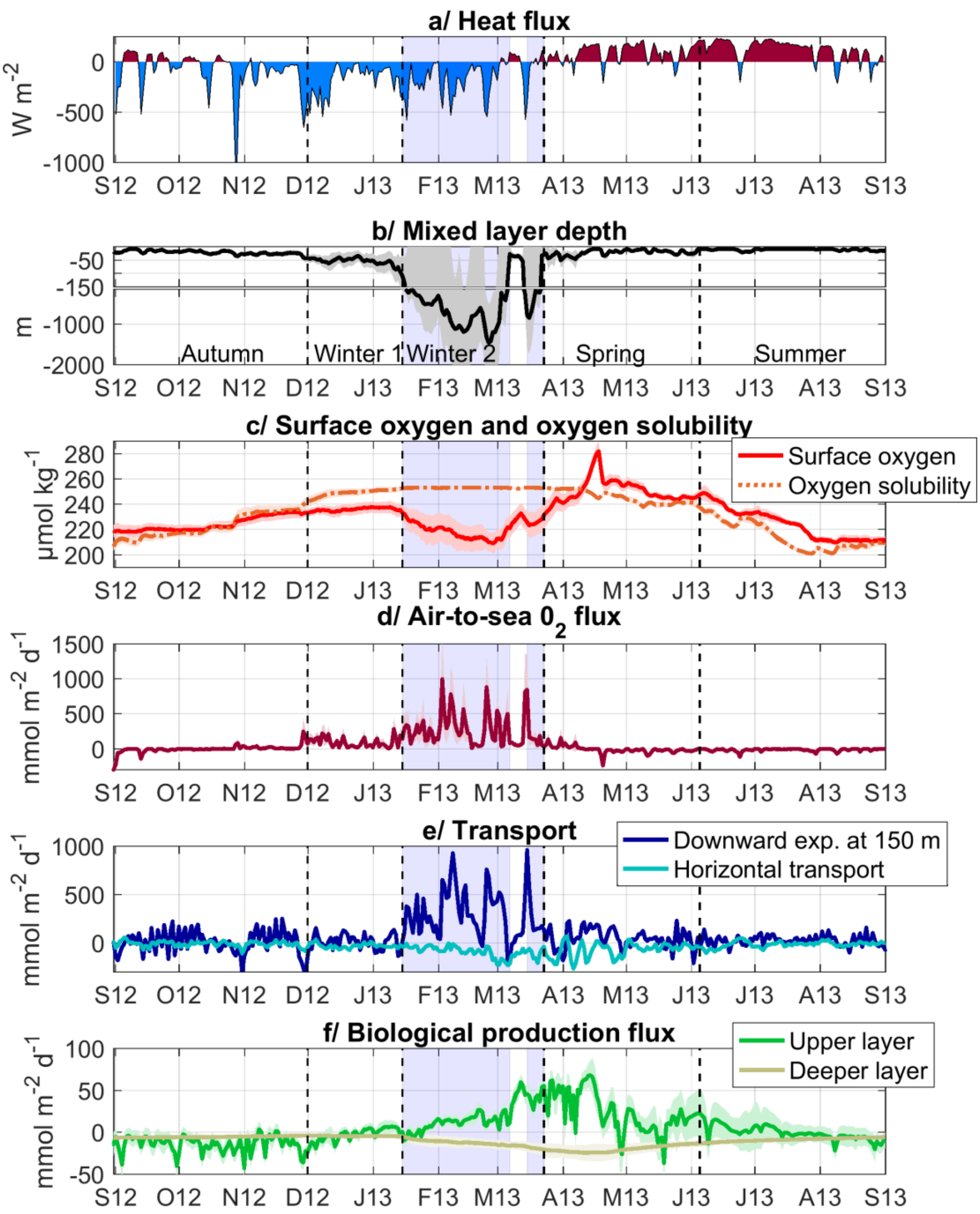

**Figure 6.** Time series of spatially averaged over the convection area (spatial mean in solid line and shaded area for standard deviation), modelled (a) total heat fluxes (W m$^{-2}$), (b) mixed layer depth (m), (c) surface oxygen and oxygen solubility (µmol kg$^{-1}$), (d) air-to-sea oxygen fluxes (mmol m$^{-2}$ day$^{-1}$), (e) downward oxygen transport at 150 m (dark blue) and lateral oxygen transport towards the convection area (light blue) (mmol m$^{-2}$ day$^{-1}$), (f) biological oxygen production (see Eq. 1) (mmol m$^{-2}$ day$^{-1}$). Sources: ECMWF for heat fluxes, SYMPHONIE/Eco3M-S for the other parameters and fluxes. Blue shaded area corresponds to the deep convection period (period when spatial averaged MLD >100 m). Note that the range of y-axis varies for the different oxygen fluxes and that due to higher values, standard
deviation for vertical and lateral transport is not shown.

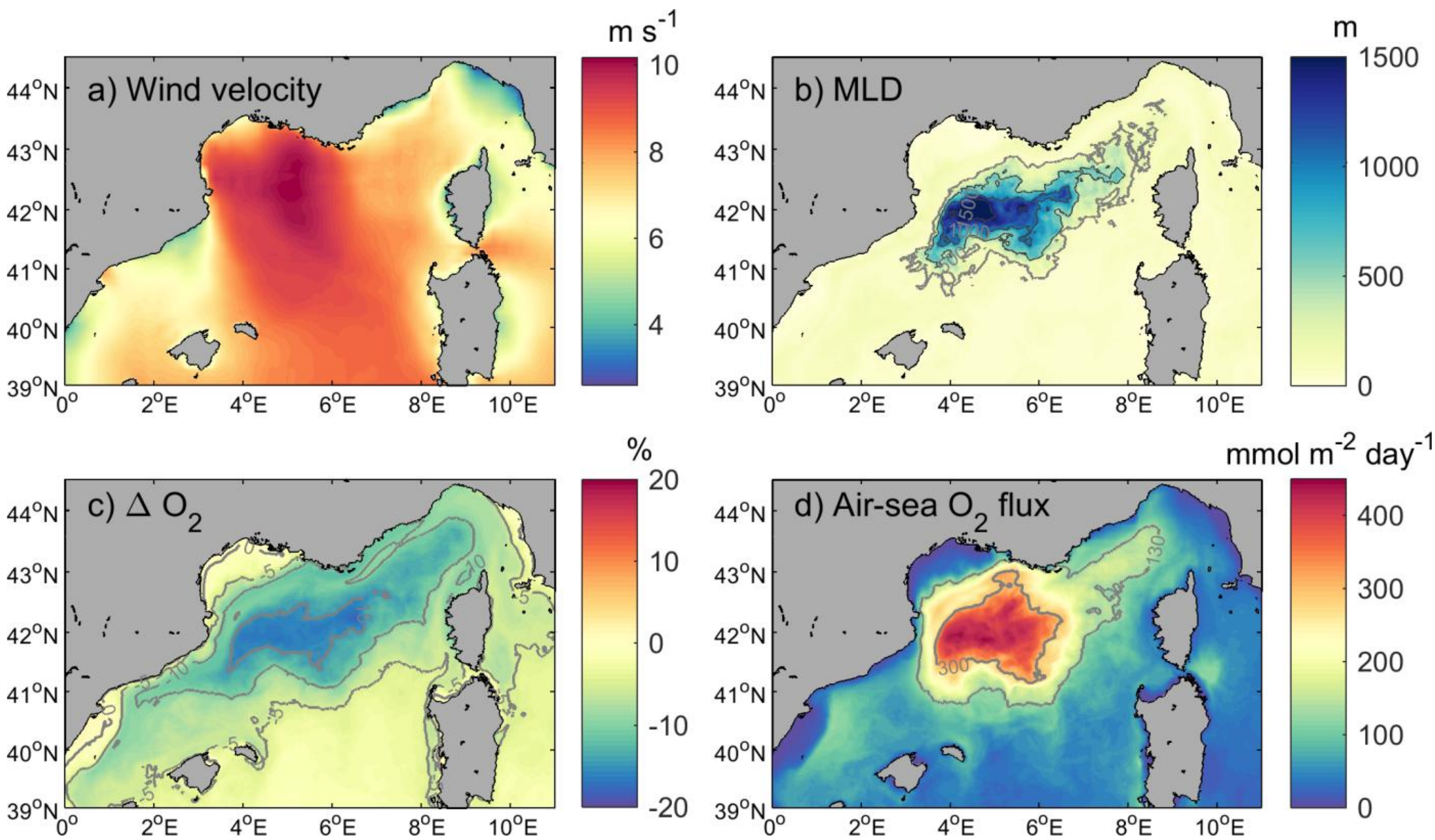

**Figure 7.** Modelled (a) wind velocity (m $s^{-1}$), (b) mixed layer depth (m) (dark grey lines represent 500, 1,000 and 1,500 isocontours and light grey line the contour of the deep convection area), (c) oxygen saturation anomaly (%) at the surface, (d) air-to-sea oxygen flux (mmol $m^{-2}$ $day^{-1}$), averaged over the 2013 deep convection period (15 January-8 March; 15 March-24 March).

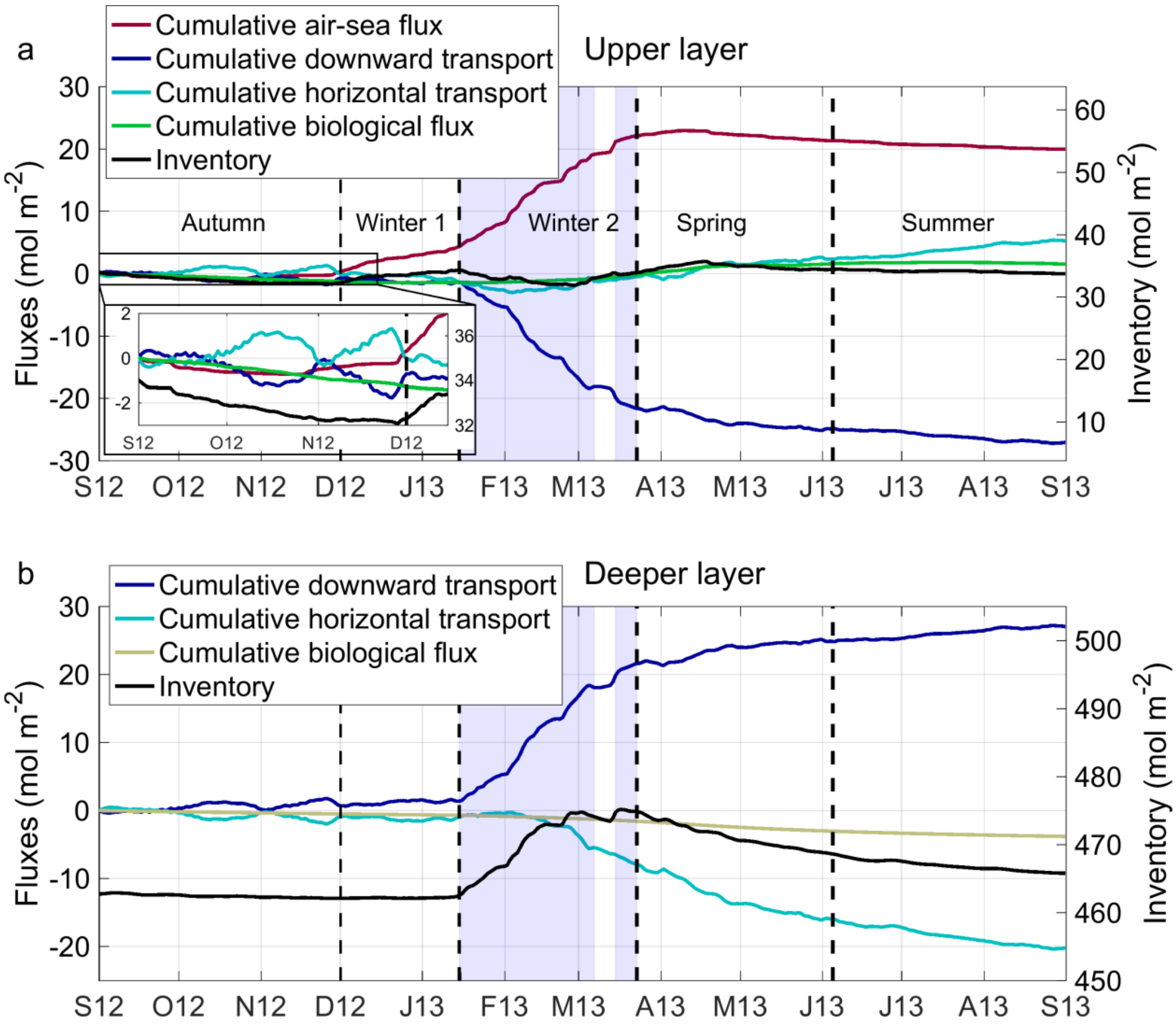

**Figure 8.** Time series from September 2012 to September 2013 of the oxygen inventory (black line) and cumulative air-sea flux (red line), downward transport (dark blue), lateral transport (positive values: input for the convection area, light blue), biological flux (green line), in the (a) upper (surface-150 m) and (b) deeper (150 m-bottom) layer. Unit: mol $m^{-2}$.

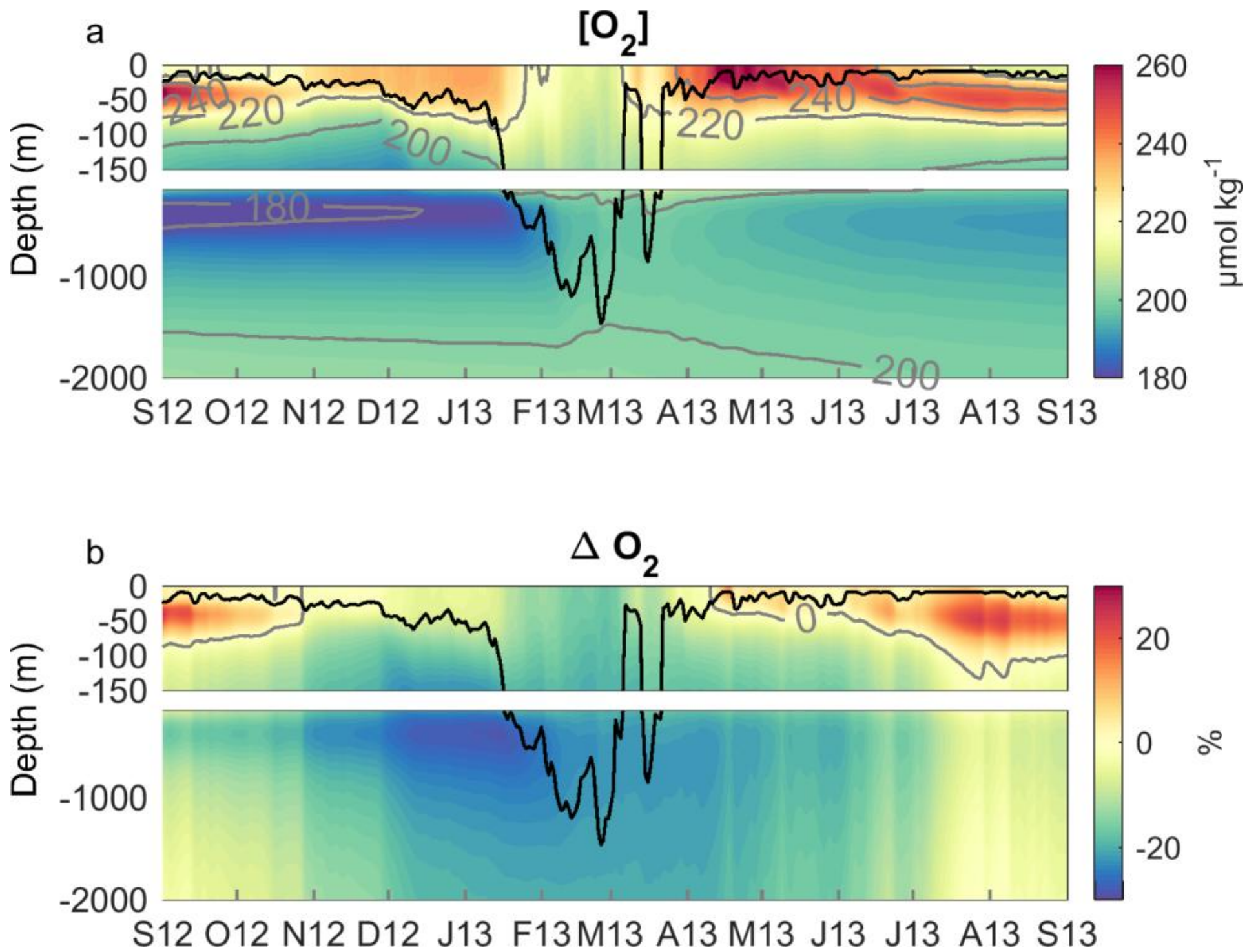

**Figure 9.** Time evolution of (a) the dissolved oxygen concentration (µmol $kg^{-1}$) and (b) the oxygen saturation anomaly (%), with mixed layer depth (m) indicated by the black line, all horizontally-averaged over the deep convection area.

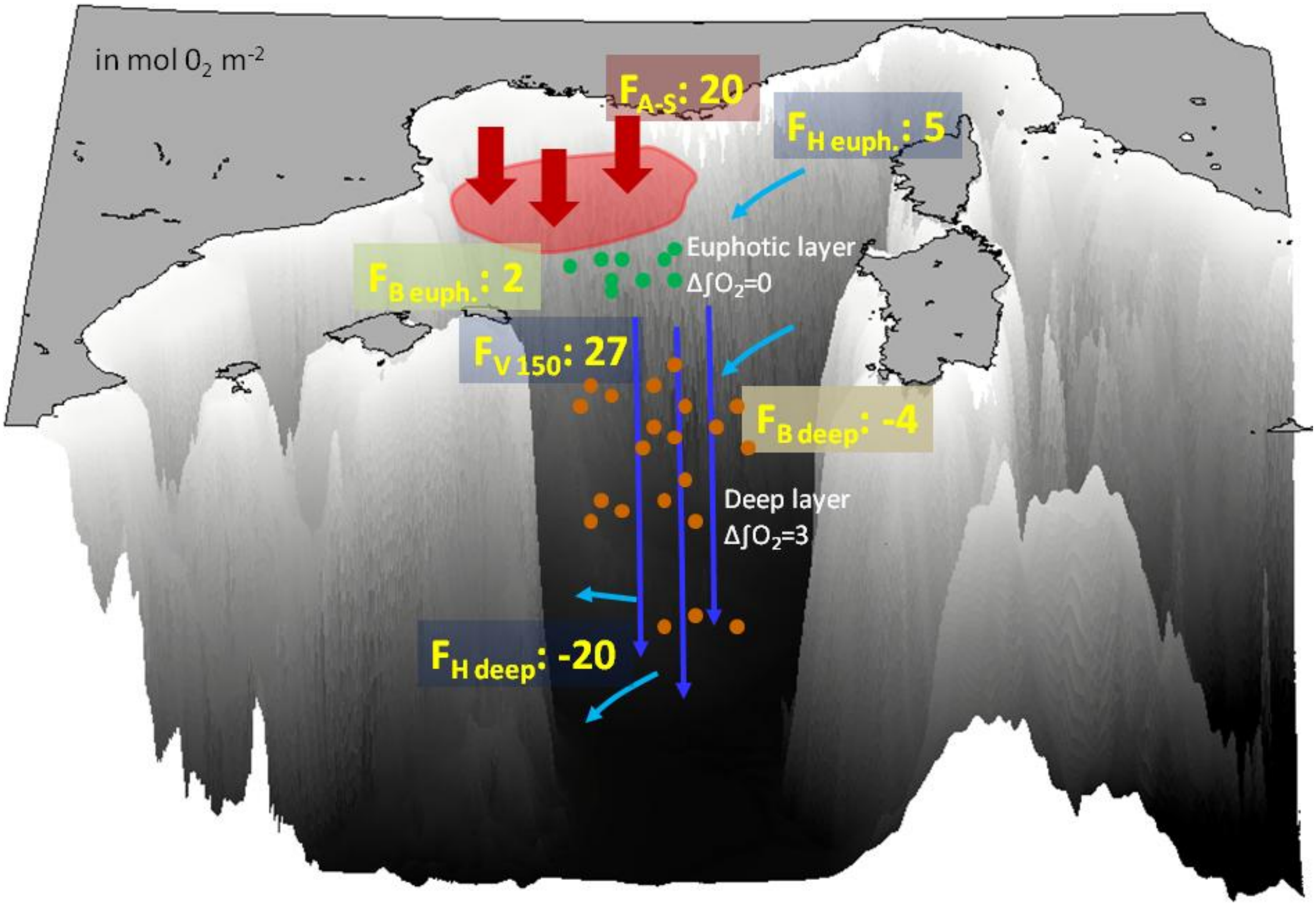

1050 **Figure 10.** Schematic showing the terms of the annual oxygen budget (in mol $O_2$ $m^{-2}$) for the north-western Mediterranean deep convection area over the period from September 2012 to September 2013. $F_{A-S}$: air to sea flux, $F_H$: net horizontal transport, $F_{v\ 150}$: net downward transport at the base of the euphotic layer (150 m), $F_B$: net biological production, $\Delta\int O_2$: variation of oxygen inventory. Positive fluxes are inputs for the deep convection zone. The terms of the budget are estimated for the upper, euphotic layer (surface-150 m), and the deeper, aphotic layers (150 m-bottom).