# Peer review of "Oxygen budget of the north-western Mediterranean deep convection region"

_Biogeosciences, 2020_

## Referee Comment (RC1) · Anonymous Referee #1 · 28 Aug 2020

Review of Manuscript "Oxygen budget for the north-western Mediterranean deep convection region" by Ulses et al. General comment to the Authors and the Editor: The ms presents an analysis, based on in situ data and model results, of the dissolved oxygen inventory of the dense water formation area in the NW-Mediterranean during one of the most active years in terms of dense water formation. They assess the inventory on a seasonal and an annual scale, describe the role of deep convection in ventilating the intermediate and deep layers of the basin, and make inferences on primary production in the euphotic layer. The ms is rigorous, very well organized, clearly written, with well-announced objectives and a logical structure that guides the reader through the author's reasoning. I recommend publication of the ms after minor revision Some more detailed comments are: Everywhere in the paper it should be written "Gulf of Lion",

not "Lions". Title: I would suggest "of" or "in the north-western" instead of "for" L35 also increased salinity reduces the solubility L40 "of marine ecosystems" L41 "implications for" L48 "subsequent density increase of surface waters" L49 "induces convective missing of surface" L56 is convection mainly responsible for this higher nutrient supply or the preconditioning given by the cyclonic circulation? In the Introduction it should be mentioned that concerning the OMZ, the Mediterranean Sea is far from what we observe in the ocean, maybe giving some numbers to exemplify L170 "Study Area" L194 instead of "Group", use "initiative" or "programme" L250 use the acronym LIW L252 move "respectively at the surface. . ..transect" at the end of the sentence L254 "During the spring cruise period" Figure 5: I could not find the explanation on why you integrate down to 1800 m and then down to 1000 m. L639 "the surface layer of the deep convection area" is the source for the intermediate and deep layer, not the convection area itself, which comprises the whole water column.

---

## Referee Comment (RC2) · Toste Tanhua (Referee) · 2 Sep 2020

The paper use a model to estimate the oxygen budget for the NW Mediterranean Sea, and use a range of relevant field data to validate the model.

The paper is very well organized and written, the logical structure is very helpful to navigate this comprehensive study. I am impressed by the overall very high quality of the paper, that covers a very relevant and significant topic. There are only a few things that could be considered in a minor revision.

The main issue is that the paper do not take the bubble effect into account. In a recent paper by Atamanchuk et al. (2020) this is discussed, and the authors conclude that " By neglecting the bubble-mediated flux component, global models may underestimate

oxygen and atmospheric potential oxygen uptake in regions of convective deep-water formation by up to an order of magnitude." I realize that this paper was published very recently, but a short discussion on the significance of bubble-mediated flux, and what it might mean for this study, would be appropriate. Minor comments: Line 34: I am not sure that reduction of deep convection related to climate change has been proven, although increased stratification etc. has . Line 56: "Massive supply of nutrients" – I guess this is by Mediterranean standards, having low nutrient concentrations in comparison to North Atlantic for instance. I agree with the statement, but maybe it needs to be put in context. Line 521: There is a recently published update of the Schneider et al 2014 paper that you could consider citing, and use as it contains data after 2012 (Li and Tanhua, 2020).

References:

Atamanchuk, D., Koelling, J., Send, U., and Wallace, D.W.R. (2020). Rapid transfer of oxygen to the deep ocean mediated by bubbles. Nature Geoscience 13, 232-237.10.1038/s41561-020-0532-2 Li, P., and Tanhua, T. (2020). Recent Changes in Deep Ventilation of the Mediterranean Sea; Evidence From Long-Term Transient Tracer Observations. Frontiers in Marine Science 7.10.3389/fmars.2020.00594

---

## Referee Comment (RC3) · Anonymous Referee #3 · 9 Sep 2020

**1   General**

The manuscript provides a detailed quantitative assessment of the preponderant contribution of dense water formation at the Gulf of Lion in the oxygenation of Mediterranean intermediate and deep waters, focusing on a particular year (Sep 2012-Sep 2013) and on the basis of high level numerical modelling (ie. coupled 3D, high resolution model).

Adding to the fact that the precise quantification of oxygen budget in this context (transport and sink/source terms) is a very timely topic (given the potential reduction of such ventilation events in the coming century), the manuscript is very well written, and succeed in handling the complexity of numerical modelling tools with accurately targeted

analyses, providing a clear and accessible result and discussion sections, as well as robust and highly relevant conclusions.

I warmly recommend the publication of the manuscript, and only report below a few minor comments or suggestions.

**2  Main Comments**

**Sect. 2.1.1**  Given the high importance of this technical aspects for the main conclusion, I would add a sentence on the diffusion and advection scheme used in Symponie (in this particular implementation).

**L150-158**  The architecture of the different model nesting and interactions, did not appeared entirely obvious to me, at first read. I would suggest a second panel to Fig1. providing a scheme of model interactions, eg. with boxes for each 4 models (NEMO, Symphonie, Basin bio, NW bio) giving temporal and spatial resolution, and mostly, arrows precising the nature of interactions (but i understand it's all offline). This is a mere suggestion to help the reader. According to the author's appreciation, an alternative would be to rework slightly this section to ensure clarity.

**Fig9, suggestion**  It seems to me that it would be relevant to add a panel to Fig. 9, indicating the biogeochemical term (VS time and depth). The vertical distribution of this term is adressed several time in the discussion, and would benefit in my opinion from a dedicated figure.

**"Biological Flux", suggestion**  As Eq.1 includes nitrification (which appears as an important component of the "biological flux", as discussed in Sect. 6.3), i wonder if it should not be called "biogeochemical flux" rather than "biological flux", in general and through the manuscript.

**L467** Something disturbs me between the sentence 463-466 and the next sentence 466-467. The first states "at the annual scale downward export below the euphotic zone ranges from 22.2 to 27.6 mol m$^{-2}$ yr$^{-1}$". The second states ,essentially, "During the convection, downward export below the euphotic zone ranges from 14.3 to 18.7 mol m$^{-2}$ yr$^{-1}$". Does the second sentence characterizes the part of the annual flux that takes place during the convection event ? Why a yr$^{-1}$ unit then ? Please clarify.

**lateral export term** It appears important to me the fact that the lateral export term in the upper layer is high, and significant in regards to atm. fluxes and local BGC net oxygen production. This indicate that the deep convection event acts as a conveyor of oxygen produced in the surface layer of surrounding areas to the deep mediterranean, and not only as a conveyor of "local oxygen". In my opinion this point should be better highlighted in the conclusions. Eventually, this aspect could be sustained with an additional panel to Fig 9, showing the vertical distribution (along time) of the lateral fluxes, but this last point is really a mere suggestion left open to the author's appreciation.

**3 Minor Comments**

**L131** $\gamma_{C/DOc} \rightarrow \gamma_{C/DOx}$

**L132** mol → mole

**L212** $y_k^o$, should be described in the previous line, with $y_k^m$. It is currently not explained.

**L213** the Root is mising in the definition of NRMSE. Also when used in the text, it is given in percentage, so maybe indicate a "100x" and "%" as is done for PB in the same line.

**L220** for readibility please favor, after the coma, "as well as modelled time evolution ... during the winter that are close to the observations".

**L224** "[The model is able to reproduce ] the deep chlorophyll maximum". Can the authors be a bit more specific , eg. the depth of the DCM, or its location, or timing or dynamics, or ..?

---

## Author Comment (AC1) · 15 Oct 2020

**Oxygen budget for the North-Western Mediterranean deep convection region**

Caroline Ulses, Claude Estournel, Marine Fourrier, Laurent Coppola, Fayçal Kessouri, Dominique Lefèvre and Patrick Marsaleix

**Responses to the comments of the anonymous Reviewer 1**

First we would like to warmly thank Reviewer 1 for his relevant and constructive comments which will help to improve the manuscript.

Answers to reviewers' comments are reported point by point. The questions and comments of the anonymous Reviewer 1 are in *blue*, the answers in black and the modifications that we propose for the revised manuscript in *black*.

Review of Manuscript "Oxygen budget for the north-western Mediterranean deep convection region" by Ulses et al. General comment to the Authors and the Editor: The ms presents an analysis, based on in situ data and model results, of the dissolved oxygen inventory of the dense water formation area in the NW-Mediterranean during one of the most active years in terms of dense water formation. They assess the inventory on a seasonal and an annual scale, describe the role of deep convection in ventilating the intermediate and deep layers of the basin, and make inferences on primary production in the euphotic layer. The ms is rigorous, very well organized, clearly written, with well-announced objectives and a logical structure that guides the reader through the author's reasoning. I recommend publication of the ms after minor revision.

Reply: We appreciate the positive assessment of Reviewer 1.

**Everywhere in the paper it should be written "Gulf of Lion", not "Lions".**

Reply: This will be corrected as suggested in the revised manuscript.

Title: I would suggest "of" or "in the north-western" instead of "for"

Reply: We will replace "for" by "of" as suggested in the title in the revised manuscript.

**L35 also increased salinity reduces the solubility**

Reply: Observational and modelling studies over the past decades show a spatial heterogeneity and a time evolution in the sign of salinity changes and trends at the global scale, with in general increases in salinity in subtropical gyres in the oceans dominated by evaporation and a freshening in regions dominated by precipitation, modulated by impacts of circulation (Durack and Wijffels, 2010). Therefore to take into account this comment we will modify the sentence as follows: "[...] to be one of the primary factors, along with the slowdown of the overturning circulation, warming-induced decrease in solubility modulated by salinity changes, and changes in C:N utilization ratios, [...]"

**L40 "of marine ecosystems"**

Reply: This will be corrected as suggested in the revised manuscript.

**L41 "implications for"**

Reply: This will be corrected as suggested in the revised manuscript.

**L48 "subsequent density increase of surface waters"**

Reply: This will be corrected as suggested in the revised manuscript.

**L49 "induces convective missing of surface"**

Reply: In the revision, we will replace "results" by "induces" as suggested.

**L56 is convection mainly responsible for this higher nutrient supply or the preconditioning given by the cyclonic circulation?**

Reply: Previous studies showed that in the north-western Mediterranean open-sea the nutrient replenishment of the surface layer essentially takes place during the deep mixing period. Using in situ profiles of nutrient at the DYFAMED station in the Ligurian Sea over the period 1995-2007, Marty and Chiavérini (2010) showed that the amount of nutrients in the surface layer is maximum during the deep convection period and that on an pluriannual scale it increased with the intensity and depth of the winter mixing (Figure 1 corresponding to Fig.9 from Marty and

Chiavérni, 2010). Also based on nutrient data at the DYFAMED station but over an extended period (1991-2011), Pasqueron de Fommervault et al. (2015) found a moderate increase of the monthly median nutrient concentrations in autumn during the preconditioning phase, from October to December (from 0.19 to 1.20 mmol m-3 for nitrate and 0.03 to 0.05 mmol m-3 for phosphate), and a strong increase in winter (between 2.60 and 2.70 mmol m-3 for nitrate and 0.11 to 0.14 mmol m-3 for phosphate). However, the observations of nutrient profiles alone do not allow deducing the vertical fluxes of nutrients, which can be more rapidly consumed by phytoplankton in autumn than during deep convection.

Figure 1. Fig. 9 extracted from Marty and Chiavérini (2010): (A) Correlation between maximum winter MLD and annual integrated chlorophyll a; (B) Correlation between maximum winter MLD and annual integrated fucoxanthin. (C) Correlation between maximum winter MLD and maximum nitrate concentration at 40 m depth in early spring.

Using a 3D physical-biogeochemical model, Ulses et al. (2016) simulated the evolution of the injection of nutrients into the surface layer due to vertical advection and mixing over the 5-year period 2004-2008 in this region. Their results showed that the nutrient vertical import was significantly correlated with the depth of the mixed layer (R=0.8, p-value

Figure 2. Fig. 10 extracted from Kessouri et al. (2017): Time series of physical and biogeochemical fluxes that impact the stock of the inorganic nitrogen and phosphorus in the surface layer (0-130 m) from September 2012 to September 2013. These fluxes are inferred from the model and averaged over the open deep convection area. (a) Net import due to vertical advection and turbulent mixing of nitrate (red) and of phosphate (blue) into the surface layer, (b) uptake of nitrate (red) and phosphate (blue), (c)

*nitrification (red) and inorganic phosphorus excretion rates (blue), and (d) ammonia excretion (red) and uptake (blue). Units: mmol*  $m^{-2} d^{-1}$ .

**In the Introduction it should be mentioned that concerning the OMZ, the Mediterranean Sea is far from what we observe in the ocean, maybe giving some numbers to exemplify**

Reply: We agree that it should be mentioned that the OMZ in the Mediterranean is much less pronounced than in the oceans where oxygen concentration is usually lower than 20  $\mu$ mol kg-1. The Mediterranean is characterized by the presence of an OML (Oxygen Minimum Layer) with oxygen concentration ranging from 170 to 180  $\mu$ mol kg-1 in the western basin (Coppola et al., 2018). Therefore we will follow the recommendation of Reviewer 1 and will add the following sentences in this Introduction:

"The oxygenation induced by recurrent deep convection together with a relatively low primary production, make the Mediterranean Sea a well oxygenated basin (Tanhua et al., 2013). In the western Mediterranean open sea, the oxygen minimum layer (OML) is located in the LIW and shows minimum oxygen concentration ranging from 170 to 180  $\mu$ mol kg-1, above ~70% of the saturation levels (Tanhua et al., 2013; Coppola et al., 2018). Thus the OML in this region is clearly less pronounced than in the open oceans or deep basins of other seas, such as the adjacent Black Sea, where hypoxic and even anoxic conditions (oxygen concentration  $<2 \text{ ml } O_2$  $l^{-1}$  or <61 µmol  $O_2$  kg-1, Diaz and Rosenberg, 2008; Breitburg et al., 2018) are encountered. However the semi-enclosed Mediterranean Sea with a fast warming was identified as one of the most vulnerable marine regions to climate change (Giorgi, 2006). Recently, regional ocean models of the Mediterranean Sea converged to predict a weakening of NW deep convection intensity under climate change scenarios by the end of the 21st century (Soto-Navarro et al., 2020). Yet, Coppola et al. (2018), by analyzing the evolution of observed oxygen profiles in the Ligurian Sea over a 20-year period, suggested that hypoxic conditions may be reached in water masses at intermediate depths after a period of 25 years without deep convection events (presuming bacterial respiration remains the same)."

**L170 "Study Area"**

Reply: This will be corrected as suggested in the revised manuscript.

**L194 instead of "Group", use "initiative" or "programme"**

Reply: This will be corrected as suggested in the revised manuscript.

**L250 use the acronym LIW**

Reply: This will be corrected as suggested in the revised manuscript.

**L252 move "respectively at the surface. . .. transect" at the end of the sentence**

Reply: This will be corrected as suggested in the revised manuscript.

**L254 "During the spring cruise period"**

Reply: This will be corrected as suggested in the revised manuscript.

**Figure 5: I could not find the explanation on why you integrate down to 1800 m and then down to 1000 m.**

Reply: We apologize for the lack of explanation on this point. For float 6901487 the data do not allow the calculation of the integrated quantity of oxygen over 1800 m due to the poor quality of the salinity data below 1000 m (Coppola et al. 2018). We therefore calculated it over 1000 m, for which we have 111/118 profiles. As we are interested in deep convection in this study, we chose to integrate the quantity of oxygen over a maximum depth, 1800 m, for the two other floats. An explanation will be added in Section 2.2.2:

"We calculated the oxygen inventory from 1800 m to the surface for floats 6901467 and 6001470 and only from 1000 m to the surface for float 6901487 due to poor quality salinity data below this depth."

**L639 "the surface layer of the deep convection area" is the source for the intermediate and deep layer, not the convection area itself, which comprises the whole water column.**

Reply: We agree with Reviewer 1. In the revised manuscript, this sentence will be modified to take into account this comment and a comment of Reviewer 3 on the role of the deep convection area of conveyor, from the surface layer to the deep layer of the western Mediterranean, of atmospheric oxygen as well as oxygen produced locally and in the surrounding areas.

Finally, we would also like to point out that we have found an error regarding the trajectory and the name of the float for which the temporal evolution of the oxygen content is shown in Figure 5b. We apologize for this error that will be corrected in the revised manuscript.

**References:**

- Durack, P. J., and S. E. Wijffels: Fifty-Year Trends in Global Ocean Salinities and Their Relationship to Broad-Scale Warming. J. Climate, 23, 4342–4362, https://doi.org/10.1175/2010JCLI3377.1, 2010
- Kessouri, F., Ulses, C., Estournel, C., Marsaleix, P., Severin, T., Pujo-Pay, M., et al.: Nitrogen and phosphorus budgets in the Northwestern Mediterranean deep convection region. Journal of Geophysical Research: Oceans, 122, 9429–9454. https://doi.org/10.1002/2016JC012665, 2017
- Marty, J. C. and J. Chiavérini. Hydrological changes in the Ligurian Sea (NW Mediterranean, DYFAMED site) during 1995–2007 and biogeochemical consequences, Biogeosci. Discuss., 7(1), 1377–1406, doi:10.5194/bgd-7-1377-2010, 2010.
- Pasqueron de Fommervault, O., C. Migon, F. D'Ortenzio, M. Ribera d'Alcal a, and L. Coppola. Temporal variability of nutrient concentrations in the northwestern Mediterranean sea (DYFAMED time-series station), Deep Sea Res., Part I, 100, 1–12, doi:10.1016/j.dsr.2015.02.006, 2015.
- Testor, P., Bosse, A., Houpert, L., Margirier, F., Mortier, L., Legoff, H., et al. Multiscale observations of deep convection in the northwestern Mediterranean Sea during winter 2012– 2013 using multiple platforms. J. Geophys. Res. Oceans 123, 1745–1776. doi: 10.1002/2016jc012671, 2018.
- Ulses, C., Auger, P.-A., Soetaert, K., Marsaleix, P., Diaz, F., Coppola, L., et al. (2016). Budget of organic carbon in the North-Western Mediterranean Open Sea over the period 2004–2008 using 3D coupled physical biogeochemical modeling. Journal of Geophysical Research: Oceans, 121, 7026–7055. https://doi.org/10.1002/2016JC011818
- Volpe, G., Nardelli, B.B., Cipollini, P., Santoleri, R., Robinson, I.S.: Seasonal to interannual phytoplankton response to physical processes in the Mediterranean Sea from satellite observations. Remote Sens. Environ. 117, 223–235, 2012.

---

## Author Comment (AC2) · 15 Oct 2020

**Oxygen budget for the North-Western Mediterranean deep convection region**

**Caroline Ulses, Claude Estournel, Marine Fourrier, Laurent Coppola, Fayçal Kessouri, Dominique Lefèvre and Patrick Marsaleix**

**Responses to the comments of Toste Tanhua, the Reviewer 2**

First we would like to warmly thank Toste Tanhua for his relevant comments and precious recommendations which will improve the manuscript.

Answers to the reviewer' comments are reported point by point. The questions and comments of Toste Tanhua are in *blue*, the answers in black and the modifications proposed in the revised manuscript in *black*.

*The paper use a model to estimate the oxygen budget for the NW Mediterranean ,and use a range of relevant field data to validate the model. The paper is very well organized and written, the logical structure is very helpful to navigate this comprehensive study. I am impressed by the overall very high quality of the paper, that covers a very relevant and significant topic. There are only a few things that could be considered in a minor revision.*

Reply: We appreciate the positive assessment of the reviewer.

*The main issue is that the paper do not take the bubble effect into account. In a recent paper by Atamanchuk et al. (2020) this is discussed, and the authors conclude that " By neglecting the bubble-mediated flux component, global models may underestimate oxygen and atmospheric potential oxygen uptake in regions of convective deep-water formation by up to an order of magnitude." I realize that this paper was published very recently, but a short discussion on the significance of bubble-mediated flux, and what it might mean for this study, would be appropriate.*

Reply: In the first version of the manuscript we presented the results of a sensitivity study on the parameterization of the oxygen flux at the air-sea interface. Two of these parameterizations, the ones proposed by Liang et al (2013) and by Bushinsky and Emerson (2018), include components of bubble-mediated fluxes. Using the parameterization proposed by Liang et al (2013) we obtained flux estimates that are in the upper range of all estimates, whereas using that of Bushinsky and Emerson (2018) the flux estimates are in the lower range. To answer more precisely this question we have performed two new sensitivity tests with the "bubble inclusive" parameterizations of Woolf (1997) (hereinafter W97) and Stanley et al (2009) (hereinafter S09), that gave ones of the best estimates in the study on the Labrador Sea by Atamanchuk et al. (2020). Both of these new tests provide estimates of

annual air-sea flux in the upper range of all estimates (20.1 mol m$^{-2}$ yr$^{-1}$ with W97 and 21.5 mol m$^{-2}$ yr$^{-1}$ with S09). In agreement with the study by Atamanchuk et al. (2020), our results with the parameterization of Stanley et al. (2009) shows, when compared to the fluxes obtained with the parameterization of Wanninkhof et al. (1992), an atmospheric oxygen uptake that started earlier in autumn, stronger fluxes in winter during peak wind periods and less outgassing in summer (Figure 1). The ratio between the two flux estimates can reach ~7 in late October, but in winter it is generally less than 2. The ratio between the two annual averages of air-sea flux, of 1.2, is less strong in our results for the northwestern Mediterranean Sea than in those obtained by Atamanchuk et al. (2020) for the Labrador Sea. This can be explained by a diffusive flux that remains significant due to the very strong undersaturation which varies between -10% and -20% on average during deep convection period over the whole studied zone (Fig. 7c of the submitted manuscript). In the Labrador Sea the undersaturation reported by Atamanchuk et al. (2020), ranging from -5% to -8%, is lower. The difference in wind intensity could also explain this difference in the ratio since this ratio reaches its largest values during days with strong wind speeds. Atamanchuk et al (2020) reported that at least 40 days during the year under study were marked by wind speeds of more than 13.8 m/s. In our study, no grid point in the zone has 40 days with wind speeds greater than 13.8 m/s. We have calculated that during the year 2012-2013 the number of days with wind speeds greater than 13.8 m/s varies between 30 and 35 days over an area representing 13% of the convection zone, located north of the central zone.

[Figure]

*Figure 1: Time series over the period September 2012-September 2013 of air-to-sea oxygen fluxes (mmol m$^{-2}$ day$^{-1}$,) estimated using the parametrizations of Stanley et al. (2009) (blue) and of Wanninkhof et al. (1992) (red), spatially averaged over the northwestern Mediterranean convection area (red).*

Finally, the estimates of annual air-sea flux (20.0 mol m$^{-2}$ yr$^{-1}$) obtained in the standard run with the "diffusive only" parameterization of Wanninkhof and McGillis (1999) are quite close to those obtained with the Stanley et al (2009) and Woolf (1997) parameterizations. This can be explained also by the strong undersaturation during the convection period and by the cubic dependence of wind speed of the Wanninkhof and McGillis (1999) parameterization. An experimental study of flux measurements in this region over an entire year would allow a better assessment of the different parameterizations.

In the revised manuscript, we will add the results of the two new sensitivity tests and references to the study of Atamanchuk et al. (2020). Modifications will therefore be made in Sections 2.1.2 (description of the biogeochemical model) and 6.1 (discussion on air-sea oxygen flux). In particular, as suggested by the Reviewer, we propose to add the following small discussion in Section 6.1:

*"Previous studies on oxygen air-sea flux in deep convection zones recommend the use of parameterizations with high transfer during periods of strong wind and convection (Copin-Montégut and Bégovic, 2002; Körtzinger et al., 2008b; Koeling et al., 2017; Atamanchuk et al., 2020). Atamanchuk et al. (2020), comparing air-sea flux estimates based on various parameterizations, found that these flux estimates may vary by an order of magnitude and warn of the possibility of a strong underestimation of oxygen air-sea flux in biogeochemical models that do not include bubble-mediated flux. In our study, the range of estimates obtained with both types of parameterizations, those that are only diffusive and those that include a bubble-mediated term, is similar. Although the parameterization of Wanninkhof and McGillis (1999) used in our standard run does not include an explicit bubble-mediated transfer term, it provides with estimates of air-sea fluxes close to those obtained with the bubble-inclusive one of Stanley et al (2009), preferred by Atamanchuk et al. (2020) in their Labrador Sea study. The strong undersaturation obtained in the north-western Mediterranean during the convection period, between -10 and -20%, may explain a greater contribution of the diffusive flux compared to the air injection by bubbles. Moreover, winter conditions are less extreme than in the Labrador Sea where strong wind speeds greater than 13.8 m/s were encountered for at least 40 days. In the north-western Mediterranean Sea, only 13% of the convection zone was characterised by a number of days with wind speeds > 13.8 m/s varying between 30 and 35 days, in winter 2013. An experimental study of flux measurements in this region over an entire year would allow a better assessment of the contribution of air injection in the total air-sea flux and hence of the use of different gas transfer parameterizations."*

*Minor comments:*

*Line 34: I am not sure that reduction of deep convection related to climate change has been proven, although increased stratification etc. has .*

Reply: Changes in circulation and convection were identified as major factors responsible for the ongoing observed and modelled deoxygenation (Plattner et al., 2002; Joos et al., 2003). The study of Brodeau and Koenigk (2016), based on an ensemble of 12 climate model

simulations shows that deep convection in the Labrador Sea started to weaken in the beginning of the twentieth in response to warming atmospheric conditions. Using observations and an ensemble of 36 simulations of CMIP5, de Lavergne et al. (2014) suggested that the activity of deep convection in the Weddell Sea was reduced under anthropogenic changes. However other reasons than climate change were also proposed by the authors to explain this decline. Therefore to take into account this comment we will modify this sentence.

*Line 56: "Massive supply of nutrients" – I guess this is by Mediterranean standards, having low nutrient concentrations in comparison to North Atlantic for instance. I agree with the statement, but maybe it needs to be put in context.*

Reply: We agree that the importance of nutrient input associated with the deep convection process should be put in the context of the oligotrophy of the Mediterranean Sea. Therefore we will modify the sentence in the revision as follows:

*"At the Mediterranean basin scale, the NW deep convection is one of the major processes responsible for an enrichment of nutrients of the euphotic layer, comparing to Atlantic influx as well as terrestrial and atmospheric inputs (Severin et al., 2014; Ulses et al., 2016; Kessouri et al., 2017). "*

*Line 521: There is a recently published update of the Schneider et al 2014 paper that you could consider citing, and use as it contains data after 2012 (Li and Tanhua, 2020).*

Reply: We thank the Reviewer for this information. The results shown in the article of Li and Tanhua (2020) complete the description of the ventilation of the western Mediterranean after 2011. We will cite these results in the discussion of our results in the revised manuscript as proposed here:

*"New oxygenated waters were also observed in the entire deep layers of the Algerian sub-basin in 2011 (Schneider et al., 2014; Stöven and Tanhua, 2015), 2014 (Keraghel et al., 2020), 2016 and 2018 (Li and Tanhua, 2020). Moreover, the results of Li and Tanhua (2020) showed a ventilation of the deep waters of the Tyrrhenian Sea through an overflow of well-oxygenated water masses from the Algerian basin into the deep layer, between 2011 and 2016."*

Finally, we would also like to point out that we have found an error regarding the trajectory and the name of the float for which the temporal evolution of the oxygen content is shown in Figure 5b. We apologize for this error that will be corrected in the revised manuscript.

**References:**

Atamanchuk, D., Koelling, J., Send, U. et al.: Rapid transfer of oxygen to the deep ocean mediated by bubbles. Nat. Geosci. 13, 232–237, https://doi.org/10.1038/s41561-020-0532-2, 2020

Brodeau, L., Koenigk, T.: Extinction of the northern oceanic deep convection in an ensemble of climate model simulations of the 20th and 21st centuries. Clim Dyn 46, 2863–2882. https://doi.org/10.1007/s00382-015-2736-5, 2016.

Joos, F., G. K. Plattner, T. F. Stocker, A. Kortzinger, and D. W. R. Wallace: Trends in marine dissolved oxygen: Implications for ocean circulation changes and the carbon budget, Eos Trans. AGU, 84(21), 197, doi:10.1029/2003EO210001, 2003.

de Lavergne, C., Palter, J., Galbraith, E. et al. Cessation of deep convection in the open Southern Ocean under anthropogenic climate change. Nature Clim Change 4, 278–282. https://doi-org.insu.bib.cnrs.fr/10.1038/nclimate2132, 2014.

Li, P., and Tanhua, T.: Recent Changes in Deep Ventilation of the Mediterranean Sea; Evidence From Long-Term Transient Tracer Observations. Frontiers in Marine Science 7.10.3389/fmars.2020.00594, 2020.

Plattner, G.-K., E. Joos, and T. F. Stocker: Revision of the global carbon budget due to changing air-sea oxygen fluxes, Global Biogeochem. Cyc, 16,1096, doi: 10.10292001GB001746, 2002.

---

## Author Comment (AC3) · 15 Oct 2020

**Oxygen budget for the North-Western Mediterranean deep convection region**

**Caroline Ulses, Claude Estournel, Marine Fourrier, Laurent Coppola, Fayçal Kessouri, Dominique Lefèvre and Patrick Marsaleix**

**Responses to the comments of the anonymous Reviewer 3**

First we would like to warmly thank the anonymous Reviewer 3 for his relevant and constructive comments which will help to improve the manuscript.

Answers to the reviewer comments are reported point by point. The questions and comments of Reviewer 3 are in *blue*, the answers in black and the modifications proposed in the revised manuscript in italic *black*.

**1 General**

*The manuscript provides a detailed quantitative assessment of the preponderant contribution of dense water formation at the Gulf of Lion in the oxygenation of Mediterranean intermediate and deep waters, focusing on a particular year (Sep 2012-Sep 2013) and on the basis of high level numerical modelling (ie. coupled 3D, high resolution model). Adding to the fact that the precise quantification of oxygen budget in this context (transport and sink/source terms) is a very timely topic (given the potential reduction of such ventilation events in the coming century), the manuscript is very well written, and succeed in handling the complexity of numerical modelling tools with accurately targeted analyses, providing a clear and accessible result and discussion sections, as well as robust and highly relevant conclusions. I warmly recommend the publication of the manuscript, and only report below a few minor comments or suggestions.*

Reply: We appreciate this positive general assessment.

**2. Main Comments**

*Sect. 2.1.1 Given the high importance of this technical aspects for the main conclusion, I would add a sentence on the diffusion and advection scheme used in Symponie (in this particular implementation).*

Reply: We agree with Reviewer 3, information on the point was missing in the manuscript. In the revised manuscript, we will specify the schemes of advection and diffusion used for this simulation in a new Sect. 2.1.3 (see response to the next comment) as follows:

*"The advection and diffusion of the biogeochemical variables were calculated using the QUICKEST (QUICK with Estimated Streaming Terms) scheme (Leonard, 1979) on the horizontal and with a centred scheme on the vertical. "*

We will also add details on diffusion in the model in Sect. 2.1.1 "The hydrodynamic model":

*"The vertical diffusion is parameterized with a prognostic equation for the turbulent kinetic energy and a diagnostic equation for the mixing and dissipation lengths, following Gaspar et al. 1990. As explained in Estournel et al. (2016a), the size of the grid is not small enough to explicitly represent convective plumes, which thus need to be parameterized. In our case, to prevent the development of static instabilities at the surface resulting in noise at the scale of the mesh, the heat and water fluxes are distributed over the whole mixed layer whose thickness is given by the depth at which the vertical density gradient becomes negative."*

*L150-158 The architecture of the different model nesting and interactions, did not appeared entirely obvious to me, at first read. I would suggest a second panel to Fig1. providing a scheme of model interactions, eg. with boxes for each 4 models (NEMO, Symphonie, Basin bio, NW bio) giving temporal and spatial resolution, and mostly, arrows precising the nature of interactions (but i understand it's all offline). This is a mere suggestion to help the reader. According to the author's appreciation, an alternative would be to rework slightly this section to ensure clarity.*

Reply: We apologize for the lack of clarity in the description of the downscaling implementation. In the revision, we will rework this section on the description of the coupled model: we will add a new sub-section 2.1.3 dedicated to the description of the particular configuration used for this study. The description of the forcing of the hydrodynamic and biogeochemical models will be transferred from Sect. 2.1.1 and 2.1.2, respectively, to this new Sect. 2.1.3. As suggested, we will make clearer the downscaling strategy at the beginning of this new Sect. 2.1.3 as proposed here:

*"A strategy of downscaling from the Mediterranean basin to the western sub-basin scale was implemented in three steps as described by Kessouri et al (2017). In a first step, the SYMPHONIE hydrodynamic model was initialized and forced at its lateral boundaries with daily analyses of the configuration PSY2V4R4 based on the NEMO ocean model at a resolution of 1/12° over the Atlantic and the Mediterranean by the Mercator-Ocean International operational system (Lellouche et al., 2013). Second, the biogeochemical model was forced at the Mediterranean basin scale by the outputs of the same NEMO simulation. In a third step, the outputs of the two previous simulations were used to initialize and force the Eco3M-S biogeochemical model over the western Mediterranean Sea."*

We are working on Figure 1 to add a second panel in this figure with explanations or a scheme of model interactions.

*Fig9, suggestion It seems to me that it would be relevant to add a panel to Fig. 9, indicating the biogeochemical term (VS time and depth). The vertical distribution of this term is adressed several time in the discussion, and would benefit in my opinion from a dedicated figure.*

Reply: For the sake of simplicity we would like to avoid adding a new panel in Figure 9. Figure 1 in this response shows the vertical distribution of the biogeochemical term. We think that the current figures are sufficient to illustrate the text and that it would be difficult to include this new figure without more comments.

[Figure]

*Figure 1: Time evolution of the net biogeochemical production of oxygen (mmol $m^{-3}yr^{-1}$)*

*"Biological Flux", suggestion As Eq.1 includes nitrification (which appears as an important component of the "biological flux", as discussed in Sect. 6.3), i wonder if it should not be called "biogeochemical flux" rather than "biological flux", in general and through the manuscript.*

Reply: In the revised manuscript, we will replace the term "biological" by "biogeochemical" in the text and in Figures 6 and 8.

*L467 Something disturbs me between the sentence 463-466 and the next sentence 466-467. The first states "at the annual scale downward export below the euphotic zone ranges from 22.2 to 27.6 mol m−2 yr−1 ". The second states ,essentially, "During the convection, downward export below the euphotic zone ranges from 14.3 to 18.7 mol m−2 yr−1 ". Does the second sentence characterizes the part of the annual flux that takes place during the convection event ? Why a yr−1 unit then ? Please clarify.*

Reply: We apologize for this error: the values correspond to the amount of oxygen in the surface layer that is exported below this layer during the convection event. We will correct this error in the revised manuscript by replacing these values with the values of downward export flux in mmol $m^{-2}$ $day^{-1}$ (as we would like to include the results of two new tests to reply to the first comment of Reviewer 2, values of export flux will change compared those in the submitted version);

*"The downward export below the euphotic zone over the deep convection period ranges from 223 to 302 mmol m$^{-2}$ day$^{-1}$ (mean value: 265 ± 30 mmol m$^{-2}$ day$^{-1}$)."*

*lateral export term It appears important to me the fact that the lateral export term in the upper layer is high, and significant in regards to atm. fluxes and local BGC net oxygen production. This indicate that the deep convection event acts as a conveyor of oxygen produced in the surface layer of surrounding areas to the deep mediterranean, and not only as a conveyor of "local oxygen". In my opinion this point should be better highlighted in the conclusions. Eventually, this aspect could be sustained with an additional panel to Fig 9, showing the vertical distribution (along time) of the lateral fluxes, but this last point is really a mere suggestion left open to the author's appreciation.*

Reply: We agree that the lateral inputs of oxygen from the surrounding areas are significant when compared to the biogeochemical production or consumption term of the budget. As suggested by the reviewer, to underline this point, we will add a discussion in the Sect. 6.2 (The role of the NW deep convection area in the ventilation of the western Mediterranean Sea) and will modify the third point of the conclusion:

In Sect. 6.2:

*"Lateral O$_2$ inputs in the surface layer occurred mainly from February to September with two peak periods, early March, a calm period between two convective events and during restratification in April. These imports are mainly related to eddies produced by the baroclinic instability that is triggered at the periphery of the convection zone when strong wind ceases (Killworth, 1976; Testor et al., 2018). These inputs from the peripheral zone contribute to the vertical export of oxygen to the aphotic layer. Firstly in the short term, the oxygen imported between two convection events is exported at depth by the following events. At longer time scales (May-September), the convection zone is also fed by the peripheral zones and in turn produces a vertical export to the aphotic layer. These exchanges are of lower intensity and concern shallower layers but are not negligible when integrated over the year. "*

In the conclusion:

*"The NW Mediterranean deep convection area acts as a conveyor of atmospheric oxygen, as well as oxygen produced locally and in the surrounding areas in the surface layer towards the intermediate and deep layer of the whole western Mediterranean Sea."*

Again, for the sake of simplicity, we would like to avoid adding a new panel in Fig. 9, considering the complexity of the processes involved in the lateral transport and the spatial heterogeneity along the boundary of the convection zone. We think that the study of the physical processes involved and the vertical and horizontal redistribution produced is beyond the scope of this paper but would justify further studies.

*3. Minor Comments*

*L131 $\gamma C/DO_c \rightarrow \gamma C/DO_x$*

Reply: This will be corrected as suggested in the revised manuscript.

*L132 mol $\rightarrow$ mole*

Reply: This will be corrected as suggested in the revised manuscript.

*L212 $yk_o$ , should be described in the previous line, with $yk_m$ . It is currently not explained.*

Reply: This will be corrected as suggested in the revised manuscript.

*L213 the Root is mising in the definition of NRMSE. Also when used in the text, it is given in percentage, so maybe indicate a "100x" and "%" as is done for PB in the same line.*

Reply: We apologize for this error. This will be addressed as suggested in the revision.

*L220 for readibility please favor, after the coma, "as well as modelled time evolution ... during the winter that are close to the observations".*

Reply: This will be corrected as suggested in the revision.

*L224 "[The model is able to reproduce ] the deep chlorophyll maximum". Can the authors be a bit more specific , eg. the depth of the DCM, or its location, or timing or dynamics, or ..?*

Reply: In the revised manuscript, we will specify what the model reproduced in the deep chlorophyll maximum as suggested:

*"These studies showed that the model is able to accurately reproduce [...] the dynamics and depth of the deep chlorophyll maximum during the stratified, oligotrophic period."*

Finally, we would also like to point out that we have found an error regarding the trajectory and the name of the float for which the temporal evolution of the oxygen content is shown in Figure 5b. We apologize for this error that will be corrected in the revised manuscript.

**References:**

Killworth, P.: The mixing and spreading phase of Medoc 1969. Progress in Oceanography, 7, 59–90, 1976.

Leonard, B.P. A stable and accurate convective modelling procedure based on quadratic upstream interpolation. Computer Methods in Applied Mechanics and Engineering, 19, 59-98, 1979.

Testor, P., Bosse, A., Houpert, L., Margirier, F., Mortier, L., Legoff, H., Dausse, D. , Labaste, M., Karstensen, J., Hayes, D., Olita, A. , Ribotti, A., Schroeder, K. , Chiggiato, J., Onken, R., Heslop, E. , Mourre, B., D'ortenzio, F. , Mayot,  N. , Lavigne, H. , de Fommervault O., Coppola, L. , Prieur, L. , Taillandier, V. , Durrieu de Madron, X. , Bourrin , F. , Many, G., Damien, P. , Estournel, C., Marsaleix, P. , Taupier-Letage, I., Raimbault, P., Waldman, R. , Bouin, M.N. , Giordani, H. , Caniaux, G. , Somot, S., Ducrocq, V. , and P. Conan (2018). Multiscale observations of deep convection in the northwestern Mediterranean Sea during winter 2012–2013 using multiple platforms. Journal of Geophysical Research: Oceans, 123. https://doi.org/10.1002/2016JC012671, 2018.

---

## Author Comment (AC4) · 15 Oct 2020

**Oxygen budget for the North-Western Mediterranean deep convection region**

**Caroline Ulses, Claude Estournel, Marine Fourrier, Laurent Coppola, Fayçal Kessouri, Dominique Lefèvre and Patrick Marsaleix**

**Responses to the comments of the anonymous Reviewer 1**

First we would like to warmly thank Reviewer 1 for his relevant and constructive comments which will help to improve the manuscript.

Answers to reviewers' comments are reported point by point. The questions and comments of the anonymous Reviewer 1 are in *blue*, the answers in black and the modifications that we propose for the revised manuscript in *black*.

*Review of Manuscript "Oxygen budget for the north-western Mediterranean deep convection region" by Ulses et al. General comment to the Authors and the Editor: The ms presents an analysis, based on in situ data and model results, of the dissolved oxygen inventory of the dense water formation area in the NW-Mediterranean during one of the most active years in terms of dense water formation. They assess the inventory on a seasonal and an annual scale, describe the role of deep convection in ventilating the intermediate and deep layers of the basin, and make inferences on primary production in the euphotic layer. The ms is rigorous, very well organized, clearly written, with well-announced objectives and a logical structure that guides the reader through the author's reasoning. I recommend publication of the ms after minor revision.*

Reply: We appreciate the positive assessment of Reviewer 1.

*Everywhere in the paper it should be written "Gulf of Lion", not "Lions".*

Reply: This will be corrected as suggested in the revised manuscript.

*Title: I would suggest "of" or "in the north-western" instead of "for"*

Reply: We will replace "for" by "of" as suggested in the title in the revised manuscript.

*L35 also increased salinity reduces the solubility*

Reply: Observational and modelling studies over the past decades show a spatial heterogeneity and a time evolution in the sign of salinity changes and trends at the global scale, with in general increases in salinity in subtropical gyres in the oceans dominated by evaporation and a freshening in regions dominated by precipitation, modulated by impacts of circulation (Durack and Wijffels, 2010). Therefore to take into account this comment we will modify the sentence as follows: *"[...] to be one of the primary factors, along with the slowdown of the overturning circulation,* **warming-induced decrease in solubility modulated by salinity changes,** *and changes in C:N utilization ratios, [...]"*

*L40 "of marine ecosystems"*

Reply: This will be corrected as suggested in the revised manuscript.

*L41 "implications for"*

Reply: This will be corrected as suggested in the revised manuscript.

*L48 "subsequent density increase of surface waters"*

Reply: This will be corrected as suggested in the revised manuscript.

*L49 "induces convective missing of surface"*

Reply: In the revision, we will replace "results" by "induces" as suggested.

*L56 is convection mainly responsible for this higher nutrient supply or the preconditioning given by the cyclonic circulation?*

Reply: Previous studies showed that in the north-western Mediterranean open-sea the nutrient replenishment of the surface layer essentially takes place during the deep mixing period. Using in situ profiles of nutrient at the DYFAMED station in the Ligurian Sea over the period 1995-2007, Marty and Chiavérini (2010) showed that the amount of nutrients in the surface layer is maximum during the deep convection period and that on an pluriannual scale it increased with the intensity and depth of the winter mixing (Figure 1 corresponding to Fig.9 from Marty and

Chiavérni, 2010). Also based on nutrient data at the DYFAMED station but over an extended period (1991-2011), Pasqueron de Fommervault et al. (2015) found a moderate increase of the monthly median nutrient concentrations in autumn during the preconditioning phase, from October to December (from 0.19 to 1.20 mmol m$^{-3}$ for nitrate and 0.03 to 0.05 mmol m$^{-3}$ for phosphate), and a strong increase in winter (between 2.60 and 2.70 mmol m$^{-3}$ for nitrate and 0.11 to 0.14 mmol m$^{-3}$ for phosphate). However, the observations of nutrient profiles alone do not allow deducing the vertical fluxes of nutrients, which can be more rapidly consumed by phytoplankton in autumn than during deep convection.

[Figure]

*Figure 1. Fig. 9 extracted from Marty and Chiavérini (2010): (A) Correlation between maximum winter MLD and annual integrated chlorophyll a; (B) Correlation between maximum winter MLD and annual integrated fucoxanthin. (C) Correlation between maximum winter MLD and maximum nitrate concentration at 40 m depth in early spring.*

Using a 3D physical-biogeochemical model, Ulses et al. (2016) simulated the evolution of the injection of nutrients into the surface layer due to vertical advection and mixing over the 5-year period 2004-2008 in this region. Their results showed that the nutrient vertical import was significantly correlated with the depth of the mixed layer (R=0.8, p-value <0.01). Kessouri et al. (2017) studied the nitrogen and phosphorus cycle in the convective zone over the same period (September 2012-September 2013) and based on the same coupled physical-biogeochemical model as in our study, they found that nutrient upward input to the surface layer remained relatively low during the preconditioning period and clearly increased during the convective period (their Figure 10A shown in this response as Figure 2).

To complete their nutrient budget and answer more precisely this question, we have calculated the vertical transport of nutrient into the surface layer during both periods using the outputs of our model: we have found that a nitrate and phosphate upward transport of 13 and 11%, respectively, of the annual upward input occurred during the preconditioning period (1 September to 15 December as defined by Testor et al., (2018)) vs 67 and 68%, respectively, during the deep convection period.

Thus it appears that a higher nutrient supply of the surface layer occurred during the convective phase than during the preconditioning phase. Obviously, the destruction of the stratification of

the water column initiated during the preconditioning influences the extension and intensity of the winter mixing and consequently those of the nutrient inputs during deep mixing as shown by Volpe et al. (2012) who studied the interannual variability of the Mediterranean ecosystem using an EOF analysis.

To take into account this comment and a comment of Reviewer 2, we will modify the sentence as follows:

*"At the Mediterranean basin scale, the NW deep convection is one of the major processes responsible for an enrichment of the euphotic layer with nutrients, comparing to Atlantic influx as well as terrestrial and atmospheric inputs (Severin et al., 2014; Ulses et al., 2016; Kessouri et al., 2017). The replenishment of the surface layer with nutrients during the deep convection is followed by an intense bloom in spring when vertical mixing weakens (Bernadello et al., 2012; Lavigne et al., 2013; Auger et al., 2014; Ulses et al., 2016; Kessouri et al., 2018)."*

[Figure]

*Figure 2. Fig. 10 extracted from Kessouri et al. (2017) : Time series of physical and biogeochemical fluxes that impact the stock of the inorganic nitrogen and phosphorus in the surface layer (0–130 m) from September 2012 to September 2013. These fluxes are inferred from the model and averaged over the open deep convection area. (a) Net import due to vertical advection and turbulent mixing of nitrate (red) and of phosphate (blue) into the surface layer, (b) uptake of nitrate (red) and phosphate (blue), (c)*

*nitrification (red) and inorganic phosphorus excretion rates (blue), and (d) ammonia excretion (red) and uptake (blue). Units: mmol m$^{-2}$ d$^{-1}$ .*

*In the Introduction it should be mentioned that concerning the OMZ, the Mediterranean Sea is far from what we observe in the ocean, maybe giving some numbers to exemplify*

Reply: We agree that it should be mentioned that the OMZ in the Mediterranean is much less pronounced than in the oceans where oxygen concentration is usually lower than 20 µmol kg$^{-1}$. The Mediterranean is characterized by the presence of an OML (Oxygen Minimum Layer) with oxygen concentration ranging from 170 to 180 µmol kg$^{-1}$ in the western basin (Coppola et al., 2018). Therefore we will follow the recommendation of Reviewer 1 and will add the following sentences in this Introduction:

*"The oxygenation induced by recurrent deep convection together with a relatively low primary production, make the Mediterranean Sea a well oxygenated basin (Tanhua et al., 2013). In the western Mediterranean open sea, the oxygen minimum layer (OML) is located in the LIW and shows minimum oxygen concentration ranging from 170 to 180 µmol kg$^{-1}$, above ~70% of the saturation levels (Tanhua et al., 2013; Coppola et al., 2018). Thus the OML in this region is clearly less pronounced than in the open oceans or deep basins of other seas, such as the adjacent Black Sea, where hypoxic and even anoxic conditions (oxygen concentration <2 ml O$_2$ l$^{-1}$ or <61 µmol O$_2$ kg$^{-1}$, Diaz and Rosenberg, 2008; Breitburg et al., 2018) are encountered. However the semi-enclosed Mediterranean Sea with a fast warming was identified as one of the most vulnerable marine regions to climate change (Giorgi, 2006). Recently, regional ocean models of the Mediterranean Sea converged to predict a weakening of NW deep convection intensity under climate change scenarios by the end of the 21$^{st}$ century (Soto-Navarro et al., 2020). Yet, Coppola et al. (2018), by analyzing the evolution of observed oxygen profiles in the Ligurian Sea over a 20-year period, suggested that hypoxic conditions may be reached in water masses at intermediate depths after a period of 25 years without deep convection events (presuming bacterial respiration remains the same)."*

*L170 "Study Area"*

Reply: This will be corrected as suggested in the revised manuscript.

*L194 instead of "Group", use "initiative" or "programme"*

Reply: This will be corrected as suggested in the revised manuscript.

*L250 use the acronym LIW*

Reply: This will be corrected as suggested in the revised manuscript.

*L252 move "respectively at the surface. . ..transect" at the end of the sentence*

Reply: This will be corrected as suggested in the revised manuscript.

*L254 "During the spring cruise period"*

Reply: This will be corrected as suggested in the revised manuscript.

*Figure 5: I could not find the explanation on why you integrate down to 1800 m and then down to 1000 m.*

Reply: We apologize for the lack of explanation on this point. For float 6901487 the data do not allow the calculation of the integrated quantity of oxygen over 1800 m due to the poor quality of the salinity data below 1000 m (Coppola et al. 2018). We therefore calculated it over 1000 m, for which we have 111/118 profiles. As we are interested in deep convection in this study, we chose to integrate the quantity of oxygen over a maximum depth, 1800 m, for the two other floats. An explanation will be added in Section 2.2.2:

*"We calculated the oxygen inventory from 1800 m to the surface for floats 6901467 and 6001470 and only from 1000 m to the surface for float 6901487 due to poor quality salinity data below this depth."*

*L639 "the surface layer of the deep convection area" is the source for the intermediate and deep layer, not the convection area itself, which comprises the whole water column.*

Reply: We agree with Reviewer 1. In the revised manuscript, this sentence will be modified to take into account this comment and a comment of Reviewer 3 on the role of the deep convection area of conveyor, from the surface layer to the deep layer of the western Mediterranean, of atmospheric oxygen as well as oxygen produced locally and in the surrounding areas.

Finally, we would also like to point out that we have found an error regarding the trajectory and the name of the float for which the temporal evolution of the oxygen content is shown in Figure 5b. We apologize for this error that will be corrected in the revised manuscript.

**References:**

Durack, P. J., and S. E. Wijffels: Fifty-Year Trends in Global Ocean Salinities and Their Relationship to Broad-Scale Warming. J. Climate, 23, 4342–4362, https://doi.org/10.1175/2010JCLI3377.1, 2010

Kessouri, F., Ulses, C., Estournel, C., Marsaleix, P., Severin, T., Pujo-Pay, M., et al.: Nitrogen and phosphorus budgets in the Northwestern Mediterranean deep convection region. Journal of Geophysical Research: Oceans, 122, 9429–9454. https://doi.org/10.1002/2016JC012665, 2017

Marty, J. C. and J. Chiavérini. Hydrological changes in the Ligurian Sea (NW Mediterranean, DYFAMED site) during 1995–2007 and biogeochemical consequences, Biogeosci. Discuss., 7(1), 1377–1406, doi:10.5194/bgd-7-1377-2010, 2010.

Pasqueron de Fommervault, O., C. Migon, F. D'Ortenzio, M. Ribera d'Alcal a, and L. Coppola. Temporal variability of nutrient concentrations in the northwestern Mediterranean sea (DYFAMED time-series station), Deep Sea Res., Part I, 100, 1–12, doi:10.1016/j.dsr.2015.02.006, 2015.

Testor, P., Bosse, A., Houpert, L., Margirier, F., Mortier, L., Legoff, H., et al. Multiscale observations of deep convection in the northwestern Mediterranean Sea during winter 2012–2013 using multiple platforms. J. Geophys. Res. Oceans 123, 1745–1776. doi: 10.1002/2016jc012671, 2018.

Ulses, C., Auger, P.-A., Soetaert, K., Marsaleix, P., Diaz, F., Coppola, L., et al. (2016). Budget of organic carbon in the North-Western Mediterranean Open Sea over the period 2004–2008 using 3D coupled physical biogeochemical modeling. Journal of Geophysical Research: Oceans, 121, 7026–7055. https://doi.org/10.1002/2016JC011818

Volpe, G., Nardelli, B.B., Cipollini, P., Santoleri, R., Robinson, I.S.: Seasonal to interannual phytoplankton response to physical processes in the Mediterranean Sea from satellite observations. Remote Sens. Environ. 117, 223–235, 2012.

---

## Author Response (AR1)

**Oxygen budget for the North-Western Mediterranean deep convection region**

Caroline Ulses, Claude Estournel, Marine Fourrier, Laurent Coppola, Fayçal Kessouri, Dominique Lefèvre and Patrick Marsaleix

**Responses to the Reviewers' comments**

Answers to reviewers' comments are reported point by point. The questions and comments of the reviewers are in *blue*, the answers in black and the modifications that we made in the revised manuscript in *black*.

**Responses to the comments of the anonymous Reviewer 1**

First we would like to warmly thank Reviewer 1 for her/his relevant and constructive comments which helped to improve the manuscript.

*Review of Manuscript "Oxygen budget for the north-western Mediterranean deep convection region" by Ulses et al. General comment to the Authors and the Editor: The ms presents an analysis, based on in situ data and model results, of the dissolved oxygen inventory of the dense water formation area in the NW-Mediterranean during one of the most active years in terms of dense water formation. They assess the inventory on a seasonal and an annual scale, describe the role of deep convection in ventilating the intermediate and deep layers of the basin, and make inferences on primary production in the euphotic layer. The ms is rigorous, very well organized, clearly written, with well-announced objectives and a logical structure that guides the reader through the author's reasoning. I recommend publication of the ms after minor revision.*

Reply: We appreciate the positive assessment of Reviewer 1.

*Everywhere in the paper it should be written "Gulf of Lion", not "Lions".*

Reply: This has been corrected as suggested in the revised manuscript.

*Title: I would suggest "of" or "in the north-western" instead of "for"*

Reply: We have replaced "for" by "of" as suggested in the title in the revised manuscript.

*L35 also increased salinity reduces the solubility*

Reply: Observational and modelling studies over the past decades show a spatial heterogeneity and a time evolution in the sign of salinity changes and trends at the global scale, with in general increases in salinity in subtropical gyres in the oceans dominated by evaporation and a freshening in regions dominated by precipitation, modulated by impacts of circulation (Durack and Wijffels, 2010). Therefore to take into account this comment we have modified the sentence as follows: *"[...] to be one of the primary factors, along with changing ventilation at intermediate depths, slowdown of the overturning circulation,* **warming-induced decrease in solubility modulated by salinity changes,** *and changes in C:N utilization ratios, [...]"*

*L40 "of marine ecosystems"*

Reply: This has been corrected as suggested in the revised manuscript.

*L41 "implications for"*

Reply: This has been corrected as suggested in the revised manuscript.

*L48 "subsequent density increase of surface waters"*

Reply: This has been corrected as suggested in the revised manuscript.

*L49 "induces convective missing of surface"*

Reply: In the revision, we have replaced "results" by "induces" as suggested.

*L56 is convection mainly responsible for this higher nutrient supply or the preconditioning given by the cyclonic circulation?*

Reply: Previous studies showed that in the north-western Mediterranean open-sea the nutrient replenishment of the surface layer essentially takes place during the deep mixing period. Using in situ profiles of nutrient at the DYFAMED station in the Ligurian Sea over the period 1995-2007, Marty and Chiavérini (2010) showed that the amount of nutrients in the surface layer is maximum during the deep convection period and that on an pluriannual scale it increased with the intensity and depth of the winter mixing (Figure 1 corresponding to Fig.9 from Marty and Chiavérni, 2010). Also based on nutrient data at the DYFAMED station but over an extended period (1991-2011), Pasqueron de Fommervault et al. (2015) found a moderate increase of the monthly median nutrient concentrations in autumn during the preconditioning phase, from October to December (from 0.19 to 1.20 mmol $m^{-3}$ for nitrate and 0.03 to

0.05 mmol m$^{-3}$ for phosphate), and a strong increase in winter (between 2.60 and 2.70 mmol m$^{-3}$ for nitrate and 0.11 to 0.14 mmol m$^{-3}$ for phosphate). However, the observations of nutrient profiles alone do not allow deducing the vertical fluxes of nutrients, which can be more rapidly consumed by phytoplankton in autumn than during deep convection.

[Figure]

*Figure 1. Fig. 9 extracted from Marty and Chiavérini (2010): (A) Correlation between maximum winter MLD and annual integrated chlorophyll a; (B) Correlation between maximum winter MLD and annual integrated fucoxanthin. (C) Correlation between maximum winter MLD and maximum nitrate concentration at 40 m depth in early spring.*

Using a 3D physical-biogeochemical model, Ulses et al. (2016) simulated the evolution of the injection of nutrients into the surface layer due to vertical advection and mixing over the 5-year period 2004-2008 in this region. Their results showed that the nutrient vertical import was significantly correlated with the depth of the mixed layer (R=0.8, p-value <0.01). Kessouri et al. (2017) studied the nitrogen and phosphorus cycle in the convective zone over the same period (September 2012-September 2013) and based on the same coupled physical-biogeochemical model as in our study, they found that nutrient upward input to the surface layer remained relatively low during the preconditioning period and clearly increased during the convective period (their Figure 10A shown in this response as Figure 2).

To complete their nutrient budget and answer more precisely this question, we have calculated the vertical transport of nutrient into the surface layer during both periods using the outputs of our model: we have found that a nitrate and phosphate upward transport of 13 and 11%, respectively, of the annual upward input occurred during the preconditioning period (1 September to 15 December as defined by Testor et al., (2018)) vs 67 and 68%, respectively, during the deep convection period.

Thus it appears that a higher nutrient supply of the surface layer occurred during the convective phase than during the preconditioning phase. Obviously, the destruction of the stratification of the water column initiated during the preconditioning influences the extension and intensity of the winter mixing and consequently those of the nutrient inputs during deep mixing as shown by Volpe et al. (2012) who studied the interannual variability of the Mediterranean ecosystem using an EOF analysis.

To take into account this comment and a comment of Reviewer 2, we have modified the sentence as follows:

*"At the Mediterranean basin scale, the deep convection occurring in the north-western region is one of the major processes responsible for an enrichment of the euphotic layer with nutrients, compared to Atlantic influx as well as terrestrial and atmospheric inputs (Severin et al., 2014; Ulses et al., 2016;*

*Kessouri et al., 2017). The replenishment of the surface layer with nutrients during the deep convection is followed by an intense bloom in spring when vertical mixing weakens (Bernadello et al., 2012; Lavigne et al., 2013; Auger et al., 2014; Ulses et al., 2016; Kessouri et al., 2018)."*

[Figure]

*Figure 2. Fig. 10 extracted from Kessouri et al. (2017) : Time series of physical and biogeochemical fluxes that impact the stock of the inorganic nitrogen and phosphorus in the surface layer (0–130 m) from September 2012 to September 2013. These fluxes are inferred from the model and averaged over the open deep convection area. (a) Net import due to vertical advection and turbulent mixing of nitrate (red) and of phosphate (blue) into the surface layer, (b) uptake of nitrate (red) and phosphate (blue), (c) nitrification (red) and inorganic phosphorus excretion rates (blue), and (d) ammonia excretion (red) and uptake (blue). Units: mmol $m^{-2}$ $d^{-1}$ .*

*In the Introduction it should be mentioned that concerning the OMZ, the Mediterranean Sea is far from what we observe in the ocean, maybe giving some numbers to exemplify*

Reply: We agree that it should be mentioned that the OMZ in the Mediterranean is much less pronounced than in the oceans where oxygen concentration is usually lower than 20 µmol $kg^{-1}$. The Mediterranean is characterized by the presence of an OML (Oxygen Minimum Layer) with oxygen concentration ranging from 170 to 180 µmol $kg^{-1}$ in the western basin (Coppola et al., 2018). Therefore we have followed the recommendation of Reviewer 1 and have added the following sentences in the Introduction:

*"The oxygenation induced by recurrent intermediate and deep convection together with a relatively low primary production, make the Mediterranean Sea a well oxygenated basin (Tanhua et al., 2013). In the western Mediterranean open sea, the oxygen minimum layer (OML) is located in the LIW and shows minimum oxygen concentration of 170-185 µmol kg$^{-1}$, above ~ 70% of the saturation levels (Tanhua et al., 2013; Coppola et al., 2018). Thus the OML in this region is clearly less pronounced than the OMZs in the open oceans or deep basins of other seas, such as the adjacent Black Sea, where hypoxic conditions (oxygen concentration <2 ml O$_2$ l$^{-1}$ or <61 µmol O$_2$ kg$^{-1}$, Diaz and Rosenberg, 2008; Breitburg et al., 2018) are encountered. However the semi-enclosed Mediterranean Sea with a fast warming was identified as one of the most vulnerable marine regions to climate change (Giorgi, 2006). Recently, regional ocean models of the Mediterranean Sea converged to predict a weakening of NW deep convection intensity under climate change scenarios by the end of the 21$^{st}$ century (Soto-Navarro et al., 2020). Yet, Coppola et al. (2018), by analyzing the evolution of observed oxygen profiles in the Ligurian Sea over a 20-year period, suggested that hypoxic conditions may be reached in water masses at intermediate depths after a period of 25 years without deep convection events (presuming bacterial respiration remains the same)."*

*L170 "Study Area"*

Reply: This has been corrected as suggested in the revised manuscript.

*L194 instead of "Group", use "initiative" or "programme"*

Reply: This has been corrected as suggested in the revised manuscript.

*L250 use the acronym LIW*

Reply: This has been corrected as suggested in the revised manuscript.

*L252 move "respectively at the surface. . ..transect" at the end of the sentence*

Reply: This has been corrected as suggested in the revised manuscript.

*L254 "During the spring cruise period"*

Reply: This has been corrected as suggested in the revised manuscript.

*Figure 5: I could not find the explanation on why you integrate down to 1800 m and then down to 1000 m.*

Reply: We apologize for the lack of explanation on this point. For float 6901487 the data do not allow the calculation of the integrated quantity of oxygen over 1800 m due to the poor quality of the salinity data below 1000 m (Coppola et al., 2017). We therefore calculated it over 1000 m, for which we have 111/118 profiles. As we are interested in deep convection in this study, we chose to integrate the quantity of oxygen over a maximum depth, 1,800 m, for the two other floats. An explanation has been added in Section 2.2.2:

*"We calculated the oxygen inventory from 1,800 m to the surface for floats 6901467 and 6001470 and only from 1,000 m to the surface for float 6901487 due to poor quality salinity data below this depth."*

*L639 "the surface layer of the deep convection area" is the source for the intermediate and deep layer, not the convection area itself, which comprises the whole water column.*

Reply: We agree with Reviewer 1. In the revised manuscript, this sentence has been modified to take into account this comment and a comment of Reviewer 3 on the role of the deep convection area of conveyor, from the surface layer to the deeper layer of the western Mediterranean, of atmospheric oxygen as well as oxygen produced both locally and in the surrounding areas:

*"The NW Mediterranean deep convection area acts as a conveyor of atmospheric oxygen, as well as of oxygen produced in the upper layer, both locally and in the surrounding areas, towards the intermediate and deep layer of the western Mediterranean Sea"*

[Figure]

Figure 1: Time series over the period September 2012-September 2013 of air-to-sea oxygen fluxes (mmol $m^{-2}$ $day^{-1,}$) estimated using the parametrizations of Stanley et al. (2009) (blue) and of Wanninkhof et al. (1992) (red), spatially averaged over the northwestern Mediterranean convection area (red).

Finally, the estimates of annual air-sea flux (20.0 mol $m^{-2}$ $yr^{-1}$) obtained in the standard run with the "diffusive only" parameterization of Wanninkhof and McGillis (1999) are quite close to those obtained with the Stanley et al (2009) and Woolf (1997) parameterizations. This can be explained also by the strong undersaturation during the convection period and by the cubic dependence of wind speed of the Wanninkhof and McGillis (1999) parameterization. An experimental study of flux measurements in this region over an entire year would allow a better assessment of the different parameterizations.

In the revised manuscript, we have added the results of the two new sensitivity tests and references to the study of Atamanchuk et al. (2020). Modifications have therefore been made in Sections 2.1.2 (description of the biogeochemical model) and 6.1 (discussion on air-sea oxygen flux). In particular, as suggested by the Reviewer, we have added the following small discussion and reference in Section 6.1:

*"Previous studies on oxygen air-sea flux in deep convection zones recommended the use of parameterizations with high transfer during periods of strong wind and convection (Copin-Montégut and Bégovic, 2002; Körtzinger et al., 2008b; Koeling et al., 2017; Atamanchuk et al., 2020). Atamanchuk et al. (2020), comparing flux estimates based on several parameterizations, found that these flux estimates may vary by an order of magnitude and warned of the possibility of a strong underestimation of air-sea*

*oxygen flux in biogeochemical models that do not include bubble-mediated terms. In our study, the range of estimates obtained with the both types of parameterizations, those that are only diffusive and those that include bubble-mediated terms, is similar. Although the parameterization of Wanninkhof and McGillis (1999) used in our standard run does not include an explicit bubble-mediated transfer term, it provides estimates of air-sea fluxes close to those obtained with the bubble-fluxes inclusive one of Stanley et al (2009), preferred by Atamanchuk et al. (2020) in their Labrador Sea study. The strong undersaturation obtained in the north-western Mediterranean during the convection period, between -10 and -20%, may explain a greater contribution of the diffusive flux compared to the air injection by bubbles. Moreover, winter conditions are less extreme than in the Labrador Sea where strong wind speeds of more than 13.8 m s$^{-1}$ were encountered for at least 40 days. In the NW Mediterranean Sea and in winter 2012/2013, only 13% of the convection area was characterised by a number of days with wind speeds > 13.8 m s$^{-1}$ varying between 30 and 35 days. An experimental study of flux measurements in this region over a whole year would allow a better assessment of the contribution of air injection in the total air-sea flux and hence of the different parameterizations of gas transfer. "*

*"Based on measurements of oxygen from a moored profiler and Argo floats and on the Stanley et al. (2009) parameterization, Atamanchuk et al. (2020) estimated an annual air-sea flux of oxygen of 19.3 ± 3.4 mol m$^{-2}$ yr$^{-1}$ for the year 2016/2017."*

*Minor comments:*

*Line 34: I am not sure that reduction of deep convection related to climate change has been proven, although increased stratification etc. has .*

Reply: Changes in circulation and convection were identified as major factors responsible for the ongoing observed and modelled deoxygenation (Plattner et al., 2002; Joos et al., 2003). The study of Brodeau and Koenigk (2016), based on an ensemble of 12 climate model simulations shows that deep convection in the Labrador Sea started to weaken in the beginning of the twentieth in response to warming atmospheric conditions. Using observations and an ensemble of 36 simulations of CMIP5, de Lavergne et al. (2014) suggested that the activity of deep convection in the Weddell Sea was reduced under anthropogenic changes. However other reasons than climate change were also proposed by the authors to explain this decline. Therefore to take into account this comment we have modified this sentence as follows:

*"Its weakening in some regions (de Lavergne et al., 2014; Brodeau and Koenigk, 2016), induced by enhanced stratification is one of the primary factors, along with changing ventilation at intermediate depths, slowdown of the overturning circulation, warming-induced decrease in solubility modulated by salinity changes and changes in C:N utilization ratios, that may explain the ongoing decline in open ocean oxygen inventory, or deoxygenation, observed and modelled since the middle of the 20$^{th}$ century (Bopp et al., 2002; Keeling and Garcia, 2002; Plattner et al., 2002; Joos et al., 2003; Keeling et al., 2010; Helm et al., 2011; Andrews et al., 2017; Ito et al., 2017; Schmidtko et al., 2017; Breitburg et al., 2018)."*

*Line 56: "Massive supply of nutrients" – I guess this is by Mediterranean standards, having low nutrient concentrations in comparison to North Atlantic for instance. I agree with the statement, but maybe it needs to be put in context.*

Reply: We agree that the importance of nutrient input associated with the deep convection process should be put in the context of the oligotrophy of the Mediterranean Sea. Therefore we have modified the sentence in the revision as follows:

*"At the Mediterranean basin scale, the deep convection occurring in the north-western region is one of the major processes responsible for an enrichment of the euphotic layer with nutrients, compared to Atlantic influx as well as terrestrial and atmospheric inputs (Severin et al., 2014; Ulses et al., 2016; Kessouri et al., 2017). "*

*Line 521: There is a recently published update of the Schneider et al 2014 paper that you could consider citing, and use as it contains data after 2012 (Li and Tanhua, 2020).*

Reply: We thank the Reviewer for this information. The results shown in the article of Li and Tanhua (2020) complete the description of the ventilation of the western Mediterranean after 2011. We have cited these results in the introduction and in the discussion of our results in the revised manuscript as follows:

*"New oxygenated waters were also observed in the entire deep layers of the Algerian sub-basin in 2011 (Schneider et al., 2014; Stöven and Tanhua, 2015), 2014 (Keraghel et al., 2020), 2016 and 2018 (Li and Tanhua, 2020). Moreover, the results of Li and Tanhua (2020) showed a ventilation of the deep waters of the Tyrrhenian Sea through an overflow of well-oxygenated water masses from the Algerian basin into the deep layer, between 2011 and 2016."*

*L150-158 The architecture of the different model nesting and interactions, did not appeared entirely obvious to me, at first read. I would suggest a second panel to Fig1. providing a scheme of model interactions, eg. with boxes for each 4 models (NEMO, Symphonie, Basin bio, NW bio) giving temporal and spatial resolution, and mostly, arrows precising the nature of interactions (but i understand it's all offline). This is a mere suggestion to help the reader. According to the author's appreciation, an alternative would be to rework slightly this section to ensure clarity.*

Reply: We apologize for the lack of clarity in the description of the downscaling implementation. In the revision, we have reworked this section on the description of the coupled model: we have added a new sub-section 2.1.3 dedicated to the description of the particular configuration used for this study. The description of the forcing of the hydrodynamic and biogeochemical models has been transferred from Sect. 2.1.1 and 2.1.2, respectively, to this new Sect. 2.1.3. As suggested, we have added sentences to clarify the downscaling strategy in this new Sect. 2.1.3 as follows:

"*A strategy of downscaling from the Mediterranean basin to the western sub-basin scale was implemented in three stages (Fig. 1a and 1c) as described by Kessouri et al (2017). In a first step, the SYMPHONIE hydrodynamic model was initialized and forced at its lateral boundaries with daily analyses of the configuration PSY2V4R4, based on the NEMO ocean model at a resolution of 1/12° over the Atlantic and the Mediterranean Sea by the Mercator-Ocean International operational system (Lellouche et al., 2013). Second, the biogeochemical model was forced at the Mediterranean basin scale by the outputs of the same NEMO simulation. In a third step, the daily outputs of the two previous simulations were used to initialize and force the Eco3M-S biogeochemical model over the western Mediterranean Sea.*"

We have reworked Figure 1 to add two new panels in this figure with a map showing the domain of the two coupled models (Fig. 1a) and a scheme of model interactions (Fig. 1c).

*Fig9, suggestion It seems to me that it would be relevant to add a panel to Fig. 9, indicating the biogeochemical term (VS time and depth). The vertical distribution of this term is adressed several time in the discussion, and would benefit in my opinion from a dedicated figure.*

Reply: For the sake of simplicity we would like to avoid adding a new panel in Figure 9. Figure 1 in this response shows the vertical distribution of the biogeochemical term. We have realized that some sentence formulations in Sect. 5.1 were awkward and confusing. We have corrected them as follows:

*Version 1:" From September to the end of November 2012 (91 days), respiration exceeded primary production throughout the water column."*

*Version 2: "From September to the end of November 2012 (91 days), depth-integrated respiration exceeded depth-integrated primary production in both upper and deeper layers (Fig. 6f)."*

*Version 1:" In the whole water column, oxygen-consuming biological processes exceeded primary production overall over this period."*

*Version 2:" Depth-integrated oxygen-consuming biogeochemical processes exceeded depth-integrated primary production on average over this period."*

We think that, with these corrections, the previous figures are sufficient to illustrate the text.

[Figure]

*Figure 1: Time evolution of the net biogeochemical production of oxygen (mmol m$^{-3}$yr$^{-1}$)*

*"Biological Flux", suggestion As Eq.1 includes nitrification (which appears as an important component of the "biological flux", as discussed in Sect. 6.3), i wonder if it should not be called "biogeochemical flux" rather than "biological flux", in general and through the manuscript.*

Reply: In the revised manuscript, we have replaced the term "biological" by "biogeochemical" in the text and in Figures 6 and 8.

*L467 Something disturbs me between the sentence 463-466 and the next sentence 466-467. The first states "at the annual scale downward export below the euphotic zone ranges from 22.2 to 27.6 mol m−2 yr−1 ". The second states ,essentially, "During the convection, downward export below the euphotic zone ranges from 14.3 to 18.7 mol m−2 yr−1 ". Does the second sentence characterizes the part of the annual flux that takes place during the convection event ? Why a yr−1 unit then ? Please clarify.*

Reply: We apologize for this error: the values correspond to the amount of oxygen in the surface layer that is exported below this layer during the convection event. We have corrected this error in the revised manuscript by replacing these values with the values of downward export flux in mmol m$^{-2}$ day$^{-1}$ (as we have included the results of two new tests to reply to the first comment of Reviewer 2, values of export flux have changed compared to those in version 1 of the manuscript):

*"The downward export below the euphotic zone over the deep convection period ranges from 223 to 302 mmol m$^{-2}$ day$^{-1}$ (mean value: 265 ± 30 mmol m$^{-2}$ day$^{-1}$)."*

*lateral export term It appears important to me the fact that the lateral export term in the upper layer is high, and significant in regards to atm. fluxes and local BGC net oxygen production. This indicate that the deep convection event acts as a conveyor of oxygen produced in the surface layer of surrounding areas to the deep mediterranean, and not only as a conveyor of "local oxygen". In my opinion this point should be better highlighted in the conclusions. Eventually, this aspect could be sustained with an additional panel to Fig 9, showing the vertical distribution (along time) of the lateral fluxes, but this last point is really a mere suggestion left open to the author's appreciation.*

Reply: We agree that the lateral inputs of oxygen from the surrounding areas are significant when compared to the biogeochemical production or consumption term of the budget. As suggested by the reviewer, to underline this point, we have added a discussion in Sect. 6.2 (The role of the NW deep convection area in the ventilation of the western Mediterranean Sea) and have modified the third point of the conclusion:

In Sect. 6.2:

*"Lateral $O_2$ inputs in the upper layer occurred mainly from February to September with two peak periods, in early March, a calm period between two convective events, and in April, during restratification. These imports were mainly related to eddies produced by the baroclinic instability that was triggered at the periphery of the convection zone when strong wind ceased (Killworth, 1976; Testor et al., 2018). These inputs from the peripheral zone contributed to the vertical export of oxygen to the aphotic layer. First in the short term, the oxygen imported between two convection events was exported at depth by the following events. At longer time scales (April-September), the convection area was also fed by the peripheral zones and in turn produced a vertical export to the aphotic layer. These exchanges were of lower intensity and concerned shallower layers but are not negligible when integrated over the year."*

In the conclusion:

*"The NW Mediterranean deep convection area acts as a conveyor of atmospheric oxygen, as well as of oxygen produced in the upper layer, both locally and in the surrounding areas, towards the intermediate and deep layer of the western Mediterranean Sea."*

Again, for the sake of simplicity, we would like to avoid adding a new panel in Fig. 9, considering the complexity of the processes involved in the lateral transport and the spatial heterogeneity along the boundary of the convection zone. We think that the study of the physical processes involved and the vertical and horizontal redistribution produced is beyond the scope of this paper but would justify further studies.

*3. Minor Comments*

*L131 γC/DOc → γC/DOx*

Reply: This has been corrected as suggested in the revised manuscript.

*L132 mol → mole*

Reply: This has been corrected as suggested in the revised manuscript.

*L212 yk o , should be described in the previous line, with yk m . It is currently not explained.*

Reply: This has been corrected as suggested in the revised manuscript.

*L213 the Root is mising in the definition of NRMSE. Also when used in the text, it is given in percentage, so maybe indicate a "100x" and "%" as is done for PB in the same line.*

Reply: We apologize for the error in the definition of NRMSE. This has been addressed as suggested in the revision.

*L220 for readibility please favor, after the coma, "as well as modelled time evolution ... during the winter that are close to the observations".*

Reply: This has been corrected as suggested in the revision.

*L224 "[The model is able to reproduce ] the deep chlorophyll maximum". Can the authors be a bit more specific , eg. the depth of the DCM, or its location, or timing or dynamics, or ..?*

Reply: In the revised manuscript, we have specified what the model reproduced in the deep chlorophyll maximum as suggested:

*"These studies showed that the model is able to accurately reproduce [...] the dynamics and depth of the deep chlorophyll maximum during the stratified, oligotrophic period."*

**Relevant changes made in the manuscript**

In addition to the corrections made on the recommendations of the reviewers, two relevant corrections have been made in version 2 of the manuscript:

- We would like to point out that we have found an error regarding the trajectory and the name of the float for which the temporal evolution of the oxygen content is shown in Figure 5b. We apologize for this error that has been corrected in the revised manuscript.

- Errors in the values and rounds of some estimates of air-sea oxygen fluxes have been corrected in the text (Sect. 6.1) and in Table 2. We apologize also for these errors which do not have a significant impact on the discussion on sensitivity tests.

[revised manuscript text omitted]